# INSIDE: LLMs' Internal States Retain the Power of Hallucination Detection

**Chao Chen**[1], **Kai Liu**[2], **Ze Chen**[1], **Yi Gu**[1], **Yue Wu**[1], **Mingyuan Tao**[1]
**Zhihang Fu**[1*], **Jieping Ye**[1*]
[1]Alibaba Cloud  [2]Zhejiang University
{ercong.cc, zhihang.fzh, yejieping}@alibaba-inc.com

## Abstract

Knowledge hallucination have raised widespread concerns for the security and reliability of deployed LLMs. Previous efforts in detecting hallucinations have been employed at logit-level uncertainty estimation or language-level self-consistency evaluation, where the semantic information is inevitably lost during the token-decoding procedure. Thus, we propose to explore the dense semantic information retained within LLMs' **IN**ternal **S**tates for halluc**I**nation **DE**tection (**INSIDE**). In particular, a simple yet effective **EigenScore** metric is proposed to better evaluate responses' self-consistency, which exploits the eigenvalues of responses' covariance matrix to measure the semantic consistency/diversity in the dense embedding space. Furthermore, from the perspective of self-consistent hallucination detection, a test time feature clipping approach is explored to truncate extreme activations in the internal states, which reduces overconfident generations and potentially benefits the detection of overconfident hallucinations. Extensive experiments and ablation studies are performed on several popular LLMs and question-answering (QA) benchmarks, showing the effectiveness of our proposal.

## 1 Introduction

Large Language Models (LLMs) have recently achieved a milestone breakthrough and demonstrated impressive abilities in various applications (Ouyang et al., 2022; OpenAI, 2023). However, it has been widely observed that even the state-of-the-art LLMs often make factually incorrect or nonsense generations (Cohen et al., 2023; Ren et al., 2022; Kuhn et al., 2022), which is also known as knowledge hallucination (Ji et al., 2023). The potentially unreliable generations make it risky to deploy LLMs in practical scenarios. Therefore, hallucination detection, that is, accurately detecting and rejecting responses when hallucinations occur in LLMs, has attracted more and more attention from the academic community (Azaria & Mitchell, 2023; Ren et al., 2022; Kuhn et al., 2022).

The token-level uncertainty estimation (e.g., predictive confidence or entropy) has shown its efficacy in hallucination detection on conventional NLP tasks (Malinin & Gales, 2020; Huang et al., 2023). However, how to derive the sentence-level uncertainty from the token-level remains a challenge, especially for modern auto-regressive LLMs whose response contents are generally diverse and sophisticated (Malinin & Gales, 2020; Kuhn et al., 2022; Duan et al., 2023). Thus, to avoid complicated token-to-sentence uncertainty derivation, researchers propose to evaluate the sentence uncertainty by the output languages directly (Kadavath et al., 2022; Yin et al., 2023; Zhou et al., 2023). Among the recent advancements, prompting LLMs to generate multiple responses to the same question and evaluating the *self-consistency* of those responses has been proven effective in hallucination detection (Wang et al., 2022; Shi et al., 2022). However, such a post-hoc semantic measurement on decoded language sentences is inferior to precisely modeling the logical consistency/divergence Manakul et al. (2023); Zhang et al. (2023).

Hence, instead of logit-level or language-level uncertainty estimation, this paper proposes to leverage the internal states of LLMs to conduct hallucination detection. The motivation is intuitive: LLMs preserve the highly-concentrated semantic information of the entire sentence within their in-

---

*Corresponding Author

ternal states (Azaria & Mitchell, 2023), allowing for the direct detection of hallucinated responses in the sentence embedding space.

In particular, with the generalized framework of **IN**ternal **S**tates for halluc**I**nation **DE**tection (**INSIDE**), this paper performs hallucination detection from two perspectives. First, skipping secondary semantic extraction via extra models, we directly measure the self-consistency/divergence of the output sentences using internal states of LLMs. In order to explore semantic consistency in the embedding space, Section 3.1 introduces an **EigenScore** metric regarding the eigenvalues of sentence embeddings' covariance matrix. Second, to handle the self-consistent (overconfident) hallucinations, we propose to rectify abnormal activations of the internal states. Specifically, Section 3.2 develops a feature clipping approach to truncate extreme features, which tends to prevent overconfident generations during the auto-regressive procedure. In Section 4, the effectiveness of our method is validated through extensive experiments on several well-established QA benchmarks.

The main contributions of our work are as follows:

- We propose a generalized INSIDE framework that leverages the internal states of LLMs to perform hallucination detection.

- We develop an EigenScore metric to measure the semantic consistency in the embedding space, and demonstrate that the proposed EigenScore represents the differential entropy in the sentence embedding space.

- A test time feature clipping approach is introduced to truncate extreme activations in the feature space, which implicitly reduces overconfident generations and helps identify the overconfident hallucinations.

- We achieve state-of-the-art hallucination detection performance on several QA benchmarks, and conduct extensive ablation studies to verify the efficacy of our method.

## 2 BACKGROUND ON HALLUCINATION DETECTION

In this work, we mainly focus on the knowledge hallucination detection of natural language generation based on LLMs, especially for Q&A task (Reddy et al., 2019; Kwiatkowski et al., 2019). Given an input context $\boldsymbol{x}$, a typical LLM (Zhang et al., 2022; Touvron et al., 2023a) parameterized with $\boldsymbol{\theta}$ is able to generate output sequences in autoregressive manner $y_t = f(\boldsymbol{x}, y_1, y_2, \cdots, y_{t-1}|\boldsymbol{\theta})$, where $\boldsymbol{y} = [y_1, y_2, \cdots, y_T]$ denotes the output sequence and $y_t$ denotes the t-$th$ output token. We denote $p(y_t|y_{<t}, \boldsymbol{x})$ the Maximum Softmax Probability (MSP) of $t$-th token. For a traditional classification model, the MSP measures the confidence level of the classification result and has been widely used as an uncertainty measure of predictions (Hendrycks & Gimpel, 2016). Therefore, for sequence generation task, a straightforward sequence uncertainty can be defined as the joint probability of different tokens, which is known as **Perplexity** (Ren et al., 2022),

$$P(\boldsymbol{y}|\boldsymbol{x}, \boldsymbol{\theta}) = -\frac{1}{T} \log \prod_t p(y_t|y_{<t}, \boldsymbol{x}) = -\frac{1}{T} \sum_t \log p(y_t|y_{<t}, \boldsymbol{x}) \tag{1}$$

As shorter sequences generally have lower perplexity, the length of the output sequence $T$ is utilized to normalize the joint probability. Since different tokens contribute differently to the semantics of the sentence (Raj et al., 2023; Duan et al., 2023), the perplexity defined by averaging token-level uncertainty cannot effectively capture the uncertainty of the entire sequence. It has been demonstrated that utilizing multiple generations for one input is beneficial to estimate the sequence-level uncertainty (Malinin & Gales, 2020; Kuhn et al., 2022; Manakul et al., 2023). We denote $\mathcal{Y} = [\boldsymbol{y}^1, \boldsymbol{y}^2, \cdots, \boldsymbol{y}^K]$ as $K$ generated responses for input context $\boldsymbol{x}$. For a given LLM, multiple responses could be easily obtained by the top-p/top-k sampling strategy during inference time (Touvron et al., 2023a; Kadavath et al., 2022). In Malinin & Gales (2020), the **Length Normalized Entropy** is proposed to measure the sequence-level uncertainty by making use of multiple generations, which is defined as

$$H(\mathcal{Y}|\boldsymbol{x}, \boldsymbol{\theta}) = -\mathbb{E}_{\boldsymbol{y} \in \mathcal{Y}} \frac{1}{T_{\boldsymbol{y}}} \sum_t \log p(y_t|y_{<t}, \boldsymbol{x}) \tag{2}$$

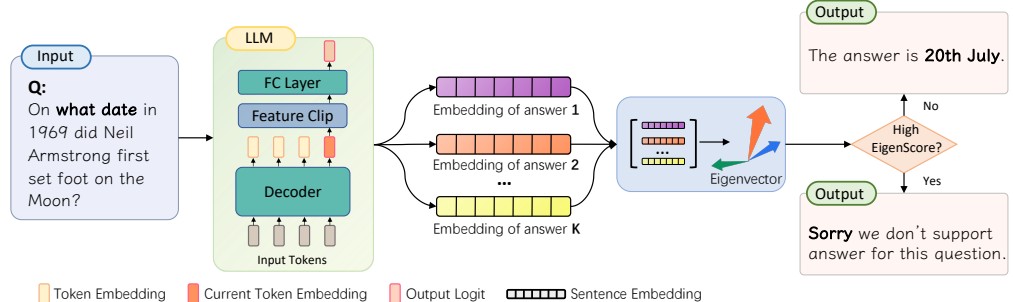

Figure 1: Illustration of our proposed hallucination detection pipeline. During inference time, for a given question, the extreme features in the penultimate layer are truncated and the EigenScore is computed based on the sentence embeddings across multiple responses.

When a model is uncertain about its response, it generates hallucination context, resulting in an answer distribution with a high entropy (Kadavath et al., 2022). It has been shown that the length-normalized entropy performs better than the non-normalized one (Lin et al., 2023).

In addition to the predictive uncertainty or entropy, the semantic consistency (Lin et al., 2023; Raj et al., 2023) among multiple responses has also been widely explored to measure the hallucination degree of LLMs, which hypothesis that the LLMs are expected to generate similar outputs if they know the input context and they are sure about the answers (Wang et al., 2022; Manakul et al., 2023). An intuitive semantic consistency metric is **Lexical Similarity** (Lin et al., 2022; 2023), which explores the average similarity across multiple answers as consistency measure

$$S(\mathcal{Y}|\boldsymbol{x}, \boldsymbol{\theta}) = \frac{1}{C} \sum_{i=1}^{K} \sum_{j=i+1}^{K} sim(\boldsymbol{y}^i, \boldsymbol{y}^j) \tag{3}$$

where $C = K \cdot (K-1)/2$ and $sim(\cdot, \cdot)$ is the similarity defined by Rouge-L Lin (2004).

## 3 METHOD

In this section, we introduce the details of our proposed INSIDE framework for hallucination detection. The whole pipeline is illustrated as Fig. 1. In section 3.1, we demonstrate a simple but effective EigenScore metric by exploring sentence-level semantics in the internal states of LLMs. In section 3.2, a test-time feature clipping approach is introduced to effectively alleviate the issue of overconfident generation, thereby aiding in the identification of self-consistent hallucinations

### 3.1 HALLUCINATION DETECTION BY EIGENSCORE

The existing uncertainty or consistency based hallucination detection metrics are exploited in the logit or language space, which neglect the dense semantic information that is retained within the internal states of LLMs. To better exploit the dense semantic information, we propose to measure the semantic divergence in the sentence embedding space. For the $t$-th output token $y_t$, we denote the hidden embedding in the $l$-th layer as $\boldsymbol{h}_t^l \in \mathbb{R}^d$, where $d$ is the dimension of the hidden embedding ($d = 4096$ for LLaMA-7B and $d = 5120$ for LLaMA-13B). According to Ren et al. (2022); Azaria & Mitchell (2023), the sentence embedding can be obtained by averaging the token embedding $\boldsymbol{z} = \frac{1}{T} \sum_{t=1}^{T} \boldsymbol{h}_t$, or taking the last token embedding as sentence embedding $\boldsymbol{z} = \boldsymbol{h}_T$. In our main experiments, we use the embedding of the last token in the middle layer as the sentence embedding, as it effectively captures the sentence semantic (Azaria & Mitchell, 2023). The comparison results of using different sentence embeddings are demonstrated in the ablation studies 4.3. For $K$ generated sequences, the covariance matrix of $K$ sentence embeddings can be computed as

$$\boldsymbol{\Sigma} = \mathbf{Z}^\top \cdot \mathbf{J}_d \cdot \mathbf{Z} \tag{4}$$

where $\mathbf{\Sigma} \in \mathbb{R}^{K \times K}$ represents the covariance matrix that captures the relationship between different sentences in the embedding space, $\mathbf{Z} = [\boldsymbol{z}_1, \boldsymbol{z}_2, \cdots, \boldsymbol{z}_K] \in \mathbb{R}^{d \times K}$ represents the embedding matrix of different sentences, $\mathbf{J}_d = \boldsymbol{I}_d - \frac{1}{d}\mathbf{1}_d\mathbf{1}_d^\top$ is the centering matrix and $\mathbf{1}_d \in \mathbb{R}^d$ is the all-one column vector. Then, the proposed EigenScore can be defined as the logarithm determinant (LogDet) of the covariance matrix,

$$E(\mathcal{Y}|\boldsymbol{x}, \boldsymbol{\theta}) = \frac{1}{K} \log \det(\mathbf{\Sigma} + \alpha \cdot \mathbf{I}_K) \tag{5}$$

Here, $\det(\mathbf{X})$ represents the determinant of matrix $\mathbf{X}$, and a small regularization term $\alpha \cdot \mathbf{I}_K$ is added to the covariance matrix to explicitly make it full rank. Since the matrix determinant can be obtained by solving the eigenvalues, the EigenScore can be computed as

$$E(\mathcal{Y}|\boldsymbol{x}, \boldsymbol{\theta}) = \frac{1}{K} \log(\prod_i \lambda_i) = \frac{1}{K} \sum_i^K \log(\lambda_i) \tag{6}$$

where $\lambda = \{\lambda_1, \lambda_2, \cdots, \lambda_K\}$ denotes the eigenvalues of the regularized covariance matrix $\mathbf{\Sigma} + \alpha \cdot \mathbf{I}$, which can be solved by Singular Value Decomposition (SVD). Eq. 6 shows that the hallucination degree of LLM's generation can be measured by the average logarithm of the eigenvalues. The conclusion is intuitive, as the eigenvalues of covariance matrix capture the divergence and correlation relationship between embeddings of different sentences. When the LLM is confident to the answers and $K$ generations have similar semantic, the sentence embeddings will be highly correlated and most eigenvalues will be close to 0. On the contrary, when the LLM is indecisive and hallucinating contents, the model will generate multiple sentences with diverse semantics leading to more significant eigenvalues. The following remark is also provided to explain why the proposed EigenScore is a good measure of knowledge hallucination.

**Remark 1. LogDet of covariance matrix represents the differential entropy in the sentence embedding space.** Differential Entropy is the natural extension of discrete Shannon Entropy $H_e(X) = -\sum_X -p(x)\log p(x)$. The differential entropy $H_{de}(X)$ in continuous space can be defined by replacing the probability function with its density function $f(x)$ and integrating over $x$, i.e., $H_{de}(X) = -\int_x f(x)\log f(x)dx$. In principle (Zhouyin & Liu, 2021), for a multivariate Gaussian distribution $X \sim N(\boldsymbol{\mu}, \mathbf{\Sigma})$, the differential entropy can be represented as

$$H_{de}(X) = \frac{1}{2}\log\det(\mathbf{\Sigma}) + \frac{d}{2}(\log 2\pi + 1) = \frac{1}{2}\sum_{i=1}^d \log \lambda_i + C \tag{7}$$

where $d$ is the dimension of variables and $C$ is a constant. Therefore, the differential entropy is determined by the eigenvalues (LogDet) of the covariance matrix.

According to **Remark 1**, the proposed EigenScore defined by Eq. 6 represents the differential entropy in the sentence embedding space, which offers valuable insight into using EigenScore as a semantic divergence measure. Compared to existing uncertainty or consistency metrics that obtained in logit or language space (Malinin & Gales, 2020; Huang et al., 2023; Lin et al., 2022), the advantages of EigenScore are: (1) It captures the semantic divergence (entropy) in the dense embedding space, which is expected to retain highly-concentrated semantic information compared to logits or languages (Reimers & Gurevych, 2019). (2) Representing semantic divergence in embedding space can effectively solve the semantic equivalence (linguistic invariances) problem (Kuhn et al., 2022) in natural language space. (3) Fine-grained semantic relationship among different responses can be exploited by using eigenvalues of covariance matrix. Therefore, through the exploration of dense semantic information in the internal states, the EigenScore is expected to outperform existing uncertainty and consistency metrics, resulting in improved hallucination detection performance.

## 3.2 TEST TIME FEATURE CLIPPING

Recent works have shown that the LLMs are subject to the risks of self-consistent (overconfident) hallucinations (Ren et al., 2022; Ji et al., 2023), which has not been considered by existing consistency based methods. Therefore, to address those failure cases caused by overconfident generation,

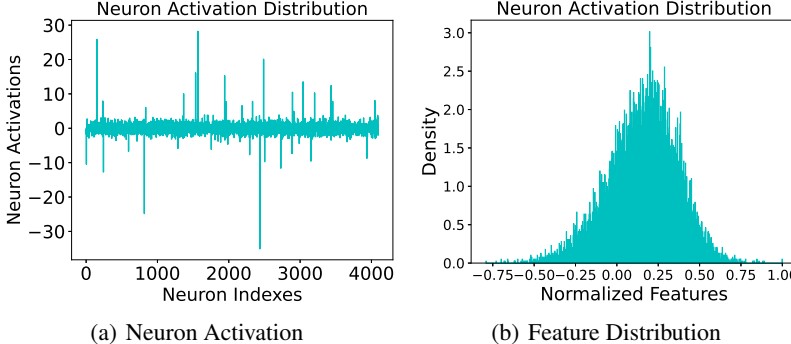

(a) Neuron Activation  (b) Feature Distribution

Figure 2: Illustration of activation distributions in the penultimate layer of LLaMA-7B. (a) Activation distribution in the penultimate layer for a randomly sampled token. (b) Activation distribution for a randomly sampled neuron activation of numerous tokens.

a test time feature clipping approach is introduced during the computation of EigenScore. As shown in Figure. 2, we illustrate the activation distribution in the penultimate layer of LLaMA-7B. An intuitive observation is that the penultimate layer of LLMs tends to exhibit numerous extreme features, consequently increasing the likelihood of generating overconfident and self-consistent generations. Inspired by prior works that rectify internal activations to reduce overconfident prediction for Out-of-Distribution (OOD) detection (Sun et al., 2021; Djurisic et al., 2022), we introduce a test time feature clipping (**FC**) method to prevent LLMs generate overconfident hallucinations. To rectify those extreme features, the FC operation is defined as the following piecewise function

$$FC(h) = \begin{cases} h_{min}, & h < h_{min} \\ h, & h_{min} \leq h \leq h_{max} \\ h_{max} & h > h_{max} \end{cases} \quad (8)$$

where $h$ represents the feature of the hidden embeddings in the penultimate layer of the LLMs, $h_{min}$ and $h_{max}$ are two thresholds for determining the minimum and maximum truncation activations. When $h_{min} = -\infty$ and $h_{max} = +\infty$, the output feature embedding is equivalent to the original output. For the determination of the optimal truncation thresholds, a memory bank which dynamically pushes and pops element in it, is utilized to conserve $N$ token embeddings during test time. Then, for each hidden neuron, the thresholds $h_{min}$ and $h_{max}$ are set to the top and bottom $p$-th percentiles of the features in the memory bank. Refer to the three-sigma-rule Pukelsheim (1994), we set $p = 0.2$ in all cases. This implies that the activations falling within the largest and smallest top 0.2% in the memory bank are identified as abnormal features and subsequently truncated for reducing overconfident generation.

## 4 EXPERIMENTS

### 4.1 EXPERIMENTAL SETUP

**Datasets.** We utilize four widely used question answering (QA) datasets for evaluation, including two open-book conversational QA datasets CoQA (Reddy et al., 2019) and SQuAD (Rajpurkar et al., 2016), as well as two closed-book QA datasets TriviaQA (Joshi et al., 2017) and Natural Questions (NQ) (Kwiatkowski et al., 2019). We follow Lin et al. (2023) to utilize the development split of CoQA with 7983 QA pairs, the validation split of NQ with 3610 QA pairs and the validation split of the TriviaQA (*rc.nocontext subset*) with 9,960 deduplicated QA pairs. For the SQuAD dataset, we filter out the QA pairs with their flag *is_impossible = True*, and utilize the subset of the development-v2.0 split with 5928 QA pairs. The lengths of the sequences vary in the four datasets. Specifically, the ground truth answers in CoQA and SQuAD are relatively longer, while and TriviaQA typically consists of answers that are only with one or two words.

**Models.** We use two representative open source LLMs, including LLaMA (Touvron et al., 2023a) and OPT (Zhang et al., 2022) in our experiments. Specifically, we consider off-the-shelf LLaMA-7B [1], LLaMA-13B [2], OPT-6.7B [3] and their corresponding tokenizer provided by Hugging Face. We use the pre-trained wights and do not finetune these models in all cases.

**Evaluation Metrics.** Following prior work Kuhn et al. (2022); Ren et al. (2022), we evaluate the hallucination detection ability of different methods by employing them to determine whether the generation is correct or not. Therefore, the area under the receiver operator characteristic curve (AUROC) and Pearson Correlation Coefficient (PCC) are utilized as the performance measure. AUROC is a popular metric to evaluate the quality of a binary classifier and uncertainty measure (Ren et al., 2022; Lin et al., 2023). Higher AUROC scores are better. PCC is utilized to measure the correlation between the hallucination detection metric and the correctness measure, which is usually defined as the ROUGE score (Lin, 2004) or semantic similarity (Reimers & Gurevych, 2019) between the generated answers and ground truth answers. A higher PCC score is better.

**Baselines.** We compare our proposal with the most popular uncertainty-based methods **Perplexity** Ren et al. (2022) and Length-normalized Entropy (**LN-Entropy**) Malinin & Gales (2020), and the consistency-based metric **Lexical Similarity** (Lin et al., 2022). Besides, in order to investigate whether traditional OOD detection methods can be used for hallucination detection, we also introduce a popular OOD detection method **Energy** score (Liu et al., 2020) as a comparison method.

**Correctness Measure.** We follow Kuhn et al. (2022); Lin et al. (2023) to utilize both the ROUGE-L (Lin, 2004) and the semantic similarity (Reimers & Gurevych, 2019) as the correctness measure. ROUGE-L [4] is an n-gram based metric that computes the longest common subsequence between two pieces of text. The generation is regarded as correct when the ROUGE-L (f-measure) is large than a given threshold, which we set to 0.5 in our main experiments. Besides, we also use the embedding similarity as the correctness measure. The sentence embeddings of model generation and the ground truth answer are extracted by the *nli-roberta-large* model [5], and the generation is regarded as true when the cosine similarity between two embeddings is larger than 0.9.

**Implementation Details.** Implementation of this work is based on pytorch and transformers libraries. For the hyperparameters that are used for sampling strategies of LLMs' decoder, we set *temperature* to 0.5, *top-p* to 0.99 and *top-k* to 5 through the experiments. The number of generations is set to $K = 10$. For the sentence embedding used in our proposal, we use the last token embedding of the sentence in the middle layer, i.e., the layer index is set to int(L/2). For the regularization term of the covariance matrix, we set $\alpha = 0.001$. For the memory bank used to conserve token embeddings, we set $N = 3000$. When implement the Energy Score, we average the token-level energy score as the sentence-level energy score.

## 4.2 MAIN RESULTS

**Effectiveness of EigenScore.** In Table. 1, we compare our proposed EigenScore with several representative reliability evaluation methods on three LLMs and four QA datasets. The results show that: (1) In both LLaMA and OPT models, our proposed EigenScore consistently outperforms other comparison methods by a large margin in CoQA, SQuAD and NQ datasets under different evaluation metrics. In particular, the EigenScore outperforms Lexical Similarity by 5.6% in CoQA and 8.9% in SQuAD with AUROC metric at most. (2) It's interesting to see that the Perplexity performs best in TriviaQA dataset but performs poorly on other datasets, especially for CoQA and SQuAD. This is because the generations and ground truth answers on TriviaQA dataset is very simple, with only one or two words in the most cases. Therefore, the performance of different methods in TriviaQA is close and by simply averaging the token-level confidence as uncertainty measure performs well. (3) On average, the performance in LLaMA-13B is better than that in LLaMA-7B and OPT-6.7B, while the performances in LLaMA-7B is slightly better than that in OPT-6.7B. It demonstrates that better hallucination detection performance can be achieved with a more powerful pre-trained LLM.

---

[1]https://huggingface.co/decapoda-research/llama-7b-hf

[2]https://huggingface.co/decapoda-research/llama-13b-hf

[3]https://huggingface.co/facebook/opt-6.7b

[4]https://github.com/google-research/google-research/tree/master/rouge

[5]https://huggingface.co/sentence-transformers/nli-roberta-large

Table 1: Hallucination detection performance evaluation of different methods on four QA tasks. AUROC (AUC) and Pearson Correlation Coefficient (PCC) are utilized to measure the performance. $AUC_s$ represents AUROC score with sentence similarity as correctness measure, and $AUC_r$ represents AUROC score with ROUGE-L score as correctness measure. All numbers are percentages.

| Models | Datasets Methods | CoQA $AUC_s$ | $AUC_r$ | PCC | SQuAD $AUC_s$ | $AUC_r$ | PCC | NQ $AUC_s$ | $AUC_r$ | PCC | TriviaQA $AUC_s$ | $AUC_r$ | PCC |
|---|---|---|---|---|---|---|---|---|---|---|---|---|---|
| | Perplexity | 64.1 | 68.3 | 20.4 | 57.5 | 60.0 | 10.2 | 74.0 | 74.7 | 30.1 | **83.6** | 83.6 | 54.4 |
| | Energy | 51.7 | 54.7 | 1.0 | 45.1 | 47.6 | -10.7 | 64.3 | 64.8 | 18.2 | 66.8 | 67.1 | 29.1 |
| LLaMA-7B | LN-Entropy | 68.7 | 73.6 | 30.6 | 70.1 | 70.9 | 30.0 | 72.8 | 73.7 | 29.8 | 83.4 | 83.2 | 54.0 |
| | Lexical Similarity | 74.8 | 77.8 | 43.5 | 74.9 | 76.4 | 44.0 | 73.8 | 75.9 | 30.6 | 82.6 | **84.0** | 55.6 |
| | **EigenScore** | **80.4** | **80.8** | **50.8** | **81.5** | **81.2** | **53.5** | **76.5** | **77.1** | **38.3** | 82.7 | 82.9 | **57.4** |
| | Perplexity | 63.2 | 66.2 | 20.1 | 59.1 | 61.7 | 14.2 | 73.5 | 73.4 | 36.3 | **84.7** | **84.5** | 56.5 |
| | Energy | 47.5 | 49.2 | -5.9 | 36.0 | 39.2 | -20.2 | 59.1 | 59.8 | 14.7 | 71.3 | 71.5 | 36.7 |
| LLaMA-13B | LN-Entropy | 68.8 | 72.9 | 31.2 | 72.4 | 74.0 | 36.6 | 74.9 | 75.2 | 39.4 | 83.4 | 83.1 | 54.2 |
| | Lexical Similarity | 74.8 | 77.6 | 44.1 | 77.4 | 79.1 | 48.6 | 74.9 | 76.8 | 40.3 | 82.9 | 84.3 | 57.5 |
| | **EigenScore** | **79.5** | **80.4** | **50.2** | **83.8** | **83.9** | **57.7** | **78.2** | **78.1** | **49.0** | 83.0 | 83.0 | **58.4** |
| | Perplexity | 60.9 | 63.5 | 11.5 | 58.4 | 69.3 | 8.6 | 76.4 | 77.0 | 32.9 | **82.6** | 82.0 | **50.0** |
| | Energy | 45.6 | 45.9 | -14.5 | 41.6 | 43.3 | -16.4 | 60.3 | 58.6 | 25.6 | 70.6 | 68.8 | 37.3 |
| OPT-6.7B | LN-Entropy | 61.4 | 65.4 | 18.0 | 65.5 | 66.3 | 22.0 | 74.0 | 76.1 | 28.4 | 79.8 | 80.0 | 43.0 |
| | Lexical Similarity | 71.2 | 74.0 | 38.4 | 72.8 | 74.0 | 39.3 | 71.5 | 74.3 | 23.1 | 78.2 | 79.7 | 42.5 |
| | **EigenScore** | **76.5** | **77.5** | **45.6** | **81.7** | **80.8** | **49.9** | **77.9** | **77.2** | **33.5** | 80.3 | 80.4 | 0.485 |

Table 2: Hallucination detection performance evaluation of different methods with and without (w/o) applying feature clipping (FC). "+FC" denotes applying feature clipping and EigenScore (w/o) denotes EigenScore without applying feature clipping. All numbers are percentages.

| Model Datasets Methods | LLaMA-7B CoQA $AUC_s$ | PCC | NQ $AUC_s$ | PCC | OPT-6.7B CoQA $AUC_s$ | PCC | NQ $AUC_s$ | PCC |
|---|---|---|---|---|---|---|---|---|
| LN-Entropy | 68.7 | 30.6 | 72.8 | 29.8 | 61.4 | 18.0 | 74.0 | 28.4 |
| LN-Entropy + FC | 70.0 | 33.4 | 73.4 | 31.1 | 62.6 | 21.4 | 74.8 | 30.3 |
| Lexical Similarity | 74.8 | 43.5 | 73.8 | 30.6 | 71.2 | 38.4 | 71.5 | 23.1 |
| Lexical Similarity + FC | 76.6 | 46.3 | 74.8 | 32.1 | 72.6 | 40.2 | 72.4 | 24.2 |
| EigenScore (w/o) | 79.3 | 48.9 | 75.9 | 38.3 | 75.3 | 43.1 | 77.1 | 32.2 |
| EigenScore | 80.4 | 50.8 | 76.5 | 38.3 | 76.5 | 45.6 | 77.9 | 33.5 |

**Effectiveness of Feature Clipping.** To demonstrate the effectiveness of the introduced test-time feature clipping, we compare the hallucination detection performance of different methods with and without applying the feature clipping technique. The results are shown in Table 2. As can be seen, the introduced feature clipping consistently improves the performance of different methods, with the largest improvement being 1.8% in AUROC.

## 4.3 ABLATION STUDIES

**Number of Generations.** For the methods that explore semantic consistency for hallucination detection, the number of generations $K$ is a key factor to the performance. Therefore, to evaluate the impact of the number of generations, we select $K$ from $\{5, 10, 15, 20, 30, 40\}$ and perform experiments with LLaMA-7B and the NQ dataset. The performance in Figure 3(a) shows that: (1) Our proposed EigenScore consistently outperforms LN-Entropy and Lexical Similarity by a large margin for different $K$. (2) When $K < 15$, the performance of different methods increases as $K$ increases and when $K > 15$, the performance tends to remain stable. The results suggeste that setting K to 20 provides the optimal trade-off between performance and inference cost. (3) Compared to EigenScore and Lexical Similarity, LN-Entropy is less sensitive to the number of generations, which demonstrates that Lexical Similarity and our EigenScore are more effective at utilizing the information in different generations.

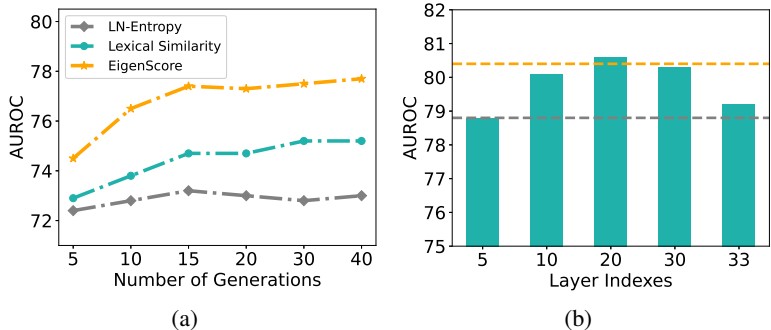

Figure 3: (a) Performance in LLaMA-7B and NQ dataset with different number of generations. (b) Performance in LLaMA-7B and CoQA dataset with sentence embedding in different layers. Orange line indicates using the last token's embedding in the middle layer (layer 17) as sentence embedding. Gray line indicates using the averaged token embedding in the last layer as sentence embedding. The performance is measured by $AUROC_s$.

Table 3: Performance evaluation with different correctness measure thresholds in LLaMA-7B and CoQA dataset. The ROUGE-L (f-measure) score and Sentence Similarity with different thresholds are employed to measure the correctness of the generated answers.

| Correctness Measures | ROUGE-L | | | Sentence Similarity | | |
|---|---|---|---|---|---|---|
| Threshold | 0.3 | 0.5 | 0.7 | 0.7 | 0.8 | 0.9 |
| Perplecity | 65.2 | 68.3 | 68.1 | 63.7 | 63.5 | 64.1 |
| LN-Entropy | 67.4 | 73.6 | 74.1 | 65.2 | 65.6 | 68.7 |
| Lexical Similarity | 75.8 | 77.8 | 79.3 | 72.8 | 73.9 | 74.8 |
| **EigenScore** | **76.4** | **80.8** | **83.5** | **75.9** | **77.2** | **80.4** |

**How EigenScore Performs with Different Sentence Embeddings.** In the main experiments, we employ the embedding of the last token in the middle layer as sentence embedding. Here, we also investigate how the model performs with different sentence embeddings. In Figure. 3(b), we show the hallucination detection performance by using sentence embedding from different layers. The results show that using the sentence embedding in the shallow and final layers yields significantly inferior performance compared to using sentence embedding in the layers close to the middle. Besides, another interesting observation is that utilizing the embedding of the last token as the sentence embedding achieves superior performance compared to simply averaging the token embeddings, which suggests that the last token of the middle layers retain more information about the truthfulness.

**Sensitivity to Correctness Measures.** It's difficult to develop automatic metrics for QA task that correlate well with human evaluations. Therefore, the choice of correctness measures is a crucial component of hallucination detection evaluation. In this section, we evaluate the performance with different correctness measure thresholds in LLaMA-7B and CoQA dataset. The experimental results are presented in Table. 3. It shows that the threshold has a great influence on the final hallucination detection performance. Significantly, our proposed EigenScore consistently outperforms comparison methods in different thresholds. Besides, the results also indicate that the hallucination detection performance of different methods will be better under a rigorous correctness measure.

**Sensitivity to Hyperparameters.** The hyperparameters, including temperature, top-k and top-p, of the LLMs' decoder determine the diversity of the generations. To evaluate the impact of those hyperparameters. We provide a sensitivity analysis in Figure 4. As observed, the performance is greatly influenced by temperature but shows little sensitivity to top-k. The performance of the consistency based methods (EigenScore and Lexical Similarity) drops significantly when the temperature is greater than 1. The optimal temperature can be selected from $[0.1, 1.0]$.

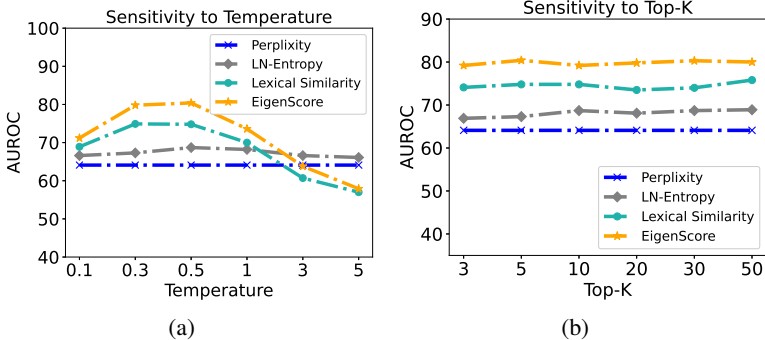

Figure 4: (a) Performance sensitivity to temperature. (b) Performance sensitivity to top-k. The performance is measured by $\text{AUROC}_s$.

## 5 RELATED WORK

**Reliability Evaluation of LLMs** During real-world deployments, the reliability of LLMs poses a substantial challenge, as LLMs reveal their propensity to exhibit unreliable generations (Ji et al., 2023; Zhang et al., 2023). Therefore, considerable efforts has been made to address the security and reliability evaluation of LLMs (Huang et al., 2023; Malinin & Gales, 2020; Kuhn et al., 2022; Kadavath et al., 2022; Cohen et al., 2023; Azaria & Mitchell, 2023). Among those methods, uncertainty based metric has been widely explored, which typically involves predictive confidence or entropy of the output token (Malinin & Gales, 2020; Kuhn et al., 2022; Duan et al., 2023). Besides, consistency based methods also play an important role in reliability evaluation, which hypothesizes that LLMs tend to generate logically inconsistent responses to the same question when they are indecisive and hallucinating contents (Kuhn et al., 2022; Raj et al., 2023; Manakul et al., 2023). Based on the consistency hypothesis, researchers also found it is feasible to prompt the LLMs to evaluate their responses themselves (Kadavath et al., 2022; Cohen et al., 2023; Manakul et al., 2023).

**Eigenvalue as Divergence Measure** The eigenvalue or determinant of covariance matrix captures the variability of the data and has been widely explored as divergence measure in a wide range of machine learning tasks (Wold et al., 1987; Kulesza & Taskar, 2011; Xu et al., 2021; Zhouyin & Liu, 2021; Cai et al., 2015). For instance, in Wold et al. (1987), the authors proposed the well-known Principal Components Analysis (PCA) and demonstrates that the most largest eigenvalues of sample covariance matrix corresponds to the principle semantic of sample set. Besides, the determinant of covariance matrix, determined by the eigenvalues, has been utilized to sample a diversity subset in determinantal point processes (DDP) (Kulesza & Taskar, 2011) and activation learning (Xu et al., 2021) tasks, which demonstrates the determinant of covariance matrix is a good diversity measure. Besides, several studies also proposed to approximate the differential entropy with the logarithm determinant of covariance matrix (Zhouyin & Liu, 2021; Klir & Wierman, 1999).

## 6 CONCLUSION

Measuring the hallucination degree of LLM's generation is of critical importance in enhancing the security and reliability of LLM-based AI systems. This work presents an INSIDE framework to exploit the semantic information that are retained within the internal states of LLMs for hallucination detection. Specifically, a simple yet effective EigenScore is proposed to measure the semantic consistency across different generations in the embedding space. Besides, to identify those self-consistent (overconfident) hallucinations which have been overlooked by previous methods, a feature clipping technique is introduced to reduce overconfident generations by truncating extreme features. Significant performance improvement has been achieved in several popular LLMs and QA benchmarks. Although our experiments focus on QA task, our method does not make any assumptions about the task modality, and we believe our method is widely applicable to other tasks, such as summarization and translation. We hope that our insights inspire future research to further explore the internal semantics of LLMs for hallucination detection.

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

Table 4: Performance comparison of different methods on TruthfulQA dataset. **LexialSim** denotes Lexical Similarity and **SelfCKGPT** denotes SelfCheckGPT. Hallucination detection accuracy is reported. **# Prompt** denotes the number of prompt templates. For ITI Li et al. (2023), we report the best number in their paper directly. All numbers are percentages.

| # Prompt | Perplexity | LN-Entropy | LexialSim | SelfCKGPT | ITI* | EigenScore |
|---|---|---|---|---|---|---|
| 5 | 70.0 | 71.2 | 73.6 | 74.2 | **83.3** | 76.7 |
| 20 | 76.4 | 77.7 | 77.9 | 76.8 | **83.3** | 79.5 |
| 50 | 73.1 | 77.9 | 73.6 | 78.3 | **83.3** | **81.3** |

## A    PERFORMANCE EVALUATION ON TRUTHFULQA

TruthfulQA is an important benchmark to evaluate the truthfulness of LLMs (Joshi et al., 2017). Therefore, we also compare our proposal with the baseline methods in the TruthfulQA benchmark. The optimal classification thresholds is determined by maximizing the **G-Mean** value, which is defined as **G-Mean** $= \sqrt{TPR * (1 - FPR)}$. The results are presented in Table 4. For the ITI Li et al. (2023), which trains multiple binary classifiers with the internal embeddings for hallucination detection, we report the best performance in their paper. As can be seen, our proposal consistently outperforms the baseline methods and achieves comparable performance as ITI when we utilize 50 in-distribution prompts. It's worth nothing that the ITI relies on training 1024 binary classifiers in TruthQA datasets, and they report the best performance (83.3) in the validation set. Therefore, their best performance is better than our proposal which has not been trained on TruthfulQA. However, training on the validation set also limits the generalization of their method on other domains (Li et al., 2023). As TruthfulQA is a very challenging dataset for LLMs, zero-shot inference results in poor performance. Therefore, we follow previous work (Bai et al., 2022) to utilize different number of in-distribution prompts during inference time. The results show that the performance could be significantly improved when we increase the number of prompts, which also explains why ITI performs good.

## B    COMPARISON WITH MORE COMPETITIVE METHODS

To demonstrate the effectiveness of our proposal, we also compare our EigenScore with several competitive methods, including **Semantic Entropy (SemanticEnt)** (Kuhn et al., 2022), Shifting Attention to Relevance (**SentSAR**) (Duan et al., 2023) and **SelfCheckGPT (SelfCKGPT)** (Manakul et al., 2023). We follow the experimental setting in Duan et al. (2023) to set the number of generation to $N = 10$ for OPT-6.7B and $N = 5$ for LLaMA. For the results of SementicEnt and SentSAR, we report the number in Duan et al. (2023) directly. For the implementation of SelfCheckGPT, we leverage the *SelfCheckBERTScore* provided in the official code package [6]. The comparison results in Table 5 demonstrate that our EigenScore significantly outperforms the competitors. Additionally, both SentSAR and SelfCheckGPT exhibit comparable performance, which is much superior to Semantic Entropy. Note that both SentSAR, SelfCheckGPT and our proposal evaluate the quality of LLMs' generation by exploring the self-consistency across multiple outputs. However, compared to Semantic Entropy (Kuhn et al., 2022) or SelfCheckGPT (Manakul et al., 2023) which relies on another language model for sentence embedding extraction, our approach leverages the internal states of LLMs, which retain highly-concentrated semantic information. Besides, the EigenScore defined by the LogDet of the sentence covariance matrix is able to capture the semantic consistency more effectively compared to the sentence-wise similarity (Manakul et al., 2023). Furthermore, the proposed feature clipping strategy allows our model to identify the overconfident hallucinations, which has not been investigated by previous works(Kuhn et al., 2022; Manakul et al., 2023)

## C    PERFORMANCE EVALUATION ON MORE LLMS

In the main experiments, we evaluate the performance of different methods in LLaMA-7B, LLaMA-13B and OPT-6.7B. To demonstrate the robustness of our method across different models, we also

---

[6]https://github.com/potsawee/selfcheckgpt

Table 5: Performance comparison of EigenScore and and several state-of-the-art methods on CoQA dataset. $AUC_s$ represents AUROC with the sentence similarity as correctness measure, and $AUC_r$ represents using ROUGE-L as correctness measure. All numbers are percentages.

| Methods | SemanticEnt | | SentSAR | | SelfCKGPT | | EigenScore | |
|---|---|---|---|---|---|---|---|---|
| | $AUC_s$ | $AUC_r$ | $AUC_s$ | $AUC_r$ | $AUC_s$ | $AUC_r$ | $AUC_s$ | $AUC_r$ |
| OPT-6.7B | 63.1 | 71.7 | 69.8 | 72.2 | 70.2 | 74.1 | **71.9** | **77.5** |
| LLaMA-7B | 64.9 | 68.2 | 70.4 | 65.8 | 68.7 | 72.9 | **71.2** | **75.7** |
| LLaMA-13B | 65.3 | 66.7 | 71.4 | 64.7 | 68.1 | 77.0 | **72.8** | **79.8** |

Table 6: Performance evaluation on LLaMA2-7B and Falcon-7B. LexicalSim denotes Lexical Similarity and SelfCKGPT denotes SelfCheckGPT. $AUC_s$ and $AUC_r$ are utilized as correctness measure. Other experimental settings are consistent with Table 1.

| Methods | | Perplecity | | LN-Entropy | | LexicalSim | | SelfCKGPT | | EigenScore | |
|---|---|---|---|---|---|---|---|---|---|---|---|
| | | $AUC_s$ | $AUC_r$ | $AUC_s$ | $AUC_r$ | $AUC_s$ | $AUC_r$ | $AUC_s$ | $AUC_r$ | $AUC_s$ | $AUC_r$ |
| LLaMA2-7b | CoQA | 62.2 | 66.6 | 69.9 | 75.2 | 74.4 | 77.5 | 72.4 | 75.1 | **78.6** | **80.7** |
| | NQ | 70.8 | 70.2 | 72.1 | 71.2 | 72.1 | 72.9 | 69.1 | 68.1 | **74.4** | **73.7** |
| Falcon-7b | CoQA | 57.0 | 60.6 | 62.6 | 63.2 | 74.8 | 76.4 | 76.7 | 77.9 | **80.8** | **80.6** |
| | NQ | 74.3 | 74.7 | 74.6 | 74.7 | 73.8 | 75.4 | 74.7 | 74.0 | **76.3** | **75.7** |

provide the performance comparison in the recent LLaMA2-7B (Touvron et al., 2023b) and Falcon-7B models (Almazrouei et al., 2023). Table 6 reveals that our proposal consistently exhibits superior performance compared to the other methods across different LLMs.

## D Computational Efficiency Analysis

As our proposal is a sampling based approach, additional inference cost is required to generate multiple outputs for accurate hallucination detection. We compare our proposal with the base LLM and other comparing methods in LLaMA-7B and LLaMA-13B. All experiments are performed on NVIDIA-A100 and we set the number of generations to $N = 10$ through the experiments. The average inference time per question is shown in Fig. 5. As observed, our EigenScore is about 10 times more efficient than the methods that rely on another large model to measure the self-consistency (such as SelfCheckGPT (Manakul et al., 2023)), and shares the similar computational overhead with the LN-Entropy and Lexical Similarity. Compared to the computational overhead of generating multiple outputs, the cost of feature clipping and EigenScore computation is negligible (0.06s). It is worth noting that the inference overhead required to generate multiple results is not linearly proportional to the time required to generate a single output, owing to the sampling and decoding strategy of the autoregressive LLM model.

## E Evaluation with Exact Match

In the main experiments, we employ the ROUGE and sentence similarity as correctness measure, which are widely used for natural language generation evaluation (Chang et al., 2023; Kuhn et al., 2022; Huang et al., 2023). In order to facilitate the comparison of our work's performance with other works, we also provide the evaluation results by employing exact match (Liang et al., 2022) as the correctness score, which is much more strict to determine a generation as correct. The results in Table 7 show similar conclusions to those in Table 1, which demonstrates that our proposal significantly outperforms the compared methods in most cases.

## F More visualization and ablation for Feature Clipping

In Fig. 6, we illustrate the distributions of neuron activation from four selected tokens. As can be seen, the distribution changes a lot across samples. Therefore, it is risky to determine the clipping

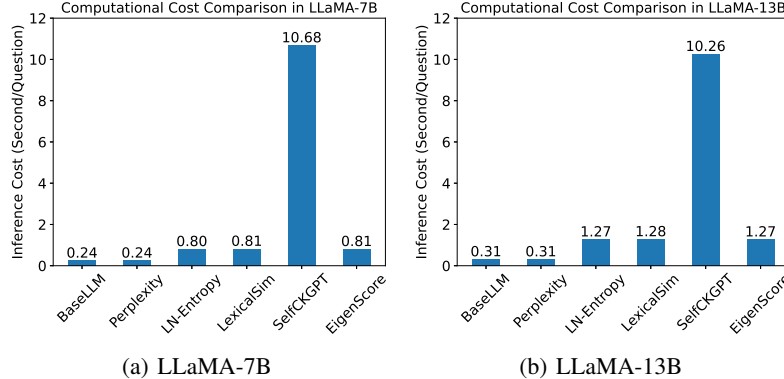

(a) LLaMA-7B                                (b) LLaMA-13B

Figure 5: Inference cost comparison of different methods in LLaMA-7B and LLaMA-13B. BaseLLM denotes the LLM without using any hallucination detection metrics. LexicalSim denotes Lexical Similarity and SelfCKGPT denotes SelfCkeckGPT.

Table 7: Performance evaluation with Exact Match as correctness measure. LexicalSim denotes the Lexical Similarity. The experimental settings are consistent with Table 1.

| Model | | Perplexity | LN-Entropy | LexicalSim | EigenScore |
|---|---|---|---|---|---|
| | CoQA | 63.7 | 70.7 | 76.1 | **83.0** |
| LLaMA-7B | SQuAD | 57.3 | 72.1 | 76.9 | **83.9** |
| | NQ | 75.3 | 75.6 | 75.8 | **80.1** |
| | TriviaQA | 82.5 | **83.4** | 81.8 | 82.4 |
| | CoQA | 59.4 | 61.7 | 71.8 | **79.4** |
| OPT-6.7B | SQuAD | 56.7 | 65.2 | 72.7 | **82.9** |
| | NQ | **79.8** | 78.1 | 73.2 | **79.8** |
| | TriviaQA | **83.8** | 81.3 | 79.3 | 82.7 |

threshold with only the current input sample (EigenScore-C). A feasible solution is to pre-compute the optimal threshold based on a batch of input samples (EigenScore-P). Besides, another solution is to dynamically record the activation values and determine the threshold during the inference process (EigenScore-MB). We have experimented with both solutions and the experimental results are presented in Table 8. The results demonstrate that determining the thresholds with a memory bank works slightly better. We attribute this variability to potential differences in the activation distributions across various datasets.

## G  LIMITATIONS AND FUTURE WORK

By exploring the internal states of LLM and introducing an EigenScore metric, the hallucination detection performance has been significantly improved. However, there are several limitations of this study. One critical limitation is that the proposed method relies on the internal information of the LLMs, therefore cannot be applied to the black-box models (OpenAI, 2023), where users can not

Table 8: Ablation study of determining the clipping threshold with different technique. EigenScore-C indicates determining the threshold with the current input sample. EigenScore-P indicates pre-computing the threshold with a batch of samples. EigenScore-MB denotes using memory bank to determine the optimal threshold. $AUC_s$ is reported.

| Methods | CoQA | NQ |
|---|---|---|
| EigenScore-C | 78.1 | 74.8 |
| EigenScore-P | 79.9 | 75.3 |
| EigenScore-MB | **80.4** | **76.5** |

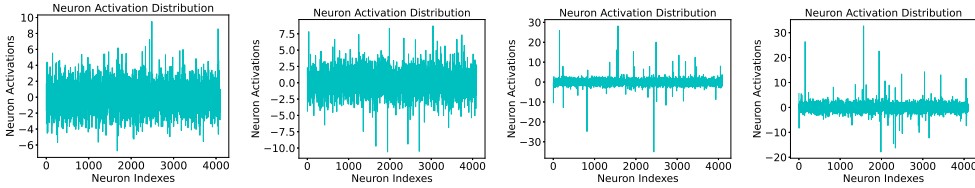

Figure 6: Activation distributions of four selected tokens in LLaMA-7B.

access the hidden states. Additionally, our proposal is a sampling-based approach, necessitating the generation of multiple outputs, which introduces additional inference overhead for accurate hallucination detection. Furthermore, another limitation is that we only focus on hallucination detection in this study and have not yet mitigate the hallucination with our proposal. In the future work, we hope to reduce the inference cost of our proposal and leverage the EigenScore to mitigate hallucinations. We believe the proposed EigenScore is a strong baseline for detecting hallucination and we invite further researchers to utilize and enhance our proposal.

## H CASES STUDY

### H.1 HALLUCINATION DETECTION CASES

We show several cases with LLaMA-7B and NQ dataset. The number of generation is set to $N = 10$. We determine the optimal detection threshold by maximizing the **G-Mean** value, which is defined as **G-Mean** $= \sqrt{TPR * (1 - FPR)}$. The hallucination detection thresholds for different metrics are Perplexity: 0.535; LN-Entropy: 0.153; LexicalSimilarity 0.489; SelfCheckGPT: 0.168; EigenScore: -1.74. For LexicalSimilarity, a score larger than the threshold indicates non-hallucination. For other metrics, a score smaller than the thresholds indicates non-hallucination. Green indicates non-hallucination answer and Red indicates hallucination answer generated by LLM. ✓ (✗) indicates that the hallucination is (not) correctly identified by the metric.

---

**Question:** the german princes who chose the holy roman empire were called
**GTAns:** prince-electors
**LLMAns:** electors
**BatchGenerations:** ['electors', 'electors', 'electors', 'electors', 'electors', 'electors', 'electors', 'electors', 'electors', 'electors']
**Perplexity**: 0.361 ✓
**LN-Entropy:** 0.027 ✓
**LexicalSimilarity:** 1.0 ✓
**SentBERTScore:** 0.0 ✓
**EigenScore:** -2.63 ✓
**EigenValue:** [4.87719579e+00 1.00000000e-03 1.00000000e-03 1.00000000e-03 1.00000000e-03 1.00000000e-03 1.00000000e-03 1.00000000e-03 1.00000000e-03 1.00000000e-03]

---

**Question:** where is fe best absorbed in the body
**GTAns:** in the duodenum
**LLMAns:** in the small intestine
**BatchGenerations:** ['in the liver', 'small intestine', 'in the intestines', 'the small intestine', 'the small intestine', 'in the liver', 'small intestine', 'fat', 'in the small intestine', 'fatty tissues']
**Perplexity**: 0.641 ✓
**LN-Entropy:** 0.213 ✓
**LexicalSimilarity:** 0.357 ✓
**SentBERTScore:** 0.258 ✓
**EigenScore:** -1.40 ✓
**EigenValue:** [3.71561676e+00 4.34496729e-01 3.77751922e-01 1.75326593e-01 9.92596975e-02 4.20723353e-02 2.49385766e-02 1.00000000e-03 1.00000000e-03 1.00000000e-03]

---

**Question:** who did the united states win its independence from
**GTAns:** the British Empire
**LLMAns:** britain
**BatchGenerations:** ['britain', 'england', 'great britain', 'great britain', 'england', 'england', 'england', 'england', 'great britain', 'great britain']
**Perplexity:** 0.598 ✗
**LN-Entropy:** 0.266 ✗
**LexicalSimilarity:** 0.415 ✗
**SentBERTScore:** 0.397 ✗
**EigenScore:** -2.23 ✓
**EigenValue:** [4.46843402e+00 2.82423429e-01 3.88702191e-02 1.00000000e-03 1.00000000e-03 1.00000000e-03 1.00000000e-03 1.00000000e-03 1.00000000e-03 1.00000000e-03]

---

**Question:** who won the most stanley cups in history
**GTAns:** Montreal Canadiens
**LLMAns:** the detroit red wings
**BatchGenerations:** ['the detroit red wings', 'the detroit red wings', 'the detroit red wings', 'the detroit red wings', 'the detroit red wings', 'the detroit red wings', 'the detroit red wings', 'the detroit red wings', 'the detroit red wings', 'the detroit red wings']
**Perplexity:** 0.366 ✗
**LN-Entropy:** 0.025 ✗
**LexicalSimilarity:** 1.0 ✗
**SentBERTScore:** 0.0 ✗
**EigenScore:** -2.63 ✗
**EigenValue:** [5.23534401e+00 1.00000000e-03 1.00000000e-03 1.00000000e-03 1.00000000e-03 1.00000000e-03 1.00000000e-03 1.00000000e-03 1.00000000e-03 1.00000000e-03]

---

**Question:** what is the second book in the alchemyst series
**GTAns:** The Magician
**LLMAns:** the alchemyst: the secret of the immortal Nicholas flamel
**BatchGenerations:** ['the magician in the middle', "the magician's nephew", 'the magician', 'the alchemyst', 'the magician', 'the alchemyst', 'the magician in the middle', 'the magician in amsterdam', 'the magician in the middle', 'the secret of the near witch']
**Perplexity:** 0.459 ✗
**LN-Entropy:** 0.207 ✓
**LexicalSimilarity:** 0.499 ✗
**SentBERTScore:** 0.431 ✓
**EigenScore:** -1.41 ✓
**EigenValue:** [2.09138499e+00 6.95605781e-01 3.85931973e-01 3.40671669e-01 2.15372994e-01 1.77304781e-01 1.00000000e-03 1.00000000e-03 1.00000000e-03 1.00000000e-03]

---

**Question:** a good that can be used in place of another good
**GTAns:** A substitute good
**LLMAns:** substitute
**BatchGenerations:** ['a substitute', 'a substitute', 'substitute', 'a substitute', 'substitute', 'substitute', 'substitute', 'substitute', 'substitute', 'substitute']
**Perplexity:** 0.614 ✗
**LN-Entropy:** 0.194 ✗
**LexicalSimilarity:** 0.844 ✓
**SentBERTScore:** 0.076 ✓
**EigenScore:** -2.40 ✓
**EigenValue:** [5.11365925e+00 1.75884104e-01 1.00000000e-03 1.00000000e-03 1.00000000e-03 1.00000000e-03 1.00000000e-03 1.00000000e-03 1.00000000e-03 1.00000000e-03]

**Question:** who said i will not go quietly into the night
**GTAns:** Welsh poet Dylan Thomas
**LLMAns:** douglas macarthur
**BatchGenerations:** ['douglas macarthur', 'elvis presley', 'douglas macarthur', 'douglas macarthur', 'douglas macarthur', 'robin williams', 'robert f. kennedy', 'robin williams', 'robert f kennedy', 'Dwight Eisenhower']
**Perplexity**: 0.502 ✗
**LN-Entropy:** 0.121 ✗
**LexicalSimilarity:** 0.178 ✓
**SentBERTScore:** 0.337 ✓
**EigenScore:** -1.61 ✓
**EigenValue:** [3.32824135e+00 5.87944819e-01 3.70390066e-01 1.70849836e-01 1.17707239e-01 5.17925563e-03 1.00000000e-03 1.00000000e-03 1.00000000e-03 1.00000000e-03]

**Question:** when did they stop saying mass in latin
**GTAns:** from 1967 onwards
**LLMAns:** 1969
**BatchGenerations:** ['in 1969', '1969', 'around 1969', 'in 1965', 'in 1969', '1969', 'In 1969, the Roman Missal was changed from Latin to the vernacular.', '1969', '1969 when the second vatican council was held', '1969']
**Perplexity**: 0.631 ✓
**LN-Entropy:** 0.152 ✗
**LexicalSimilarity:** 0.540 ✗
**SentBERTScore:** 0.273 ✓
**EigenScore:** -1.59 ✓
**EigenValue:** [3.80408192e+00 4.83987672e-01 3.03207580e-01 8.80366008e-02 6.59790286e-02 3.26742841e-02 1.00000000e-03 1.00000000e-03 1.00000000e-03 1.00000000e-03]

**Question:** where does the highway of heroes start and end
**GTAns:** Glen Miller Road in Trenton
**LLMAns:** it starts in ottawa and ends in toronto
**BatchGenerations:** ['start in ottawa and end in toronto', 'it starts in toronto and ends in ottawa', "it starts at the queen's park gate at queen's park and ends at toronto's coronation park", "it starts in toronto and ends at queen's park", 'starts in Toronto and ends in Trenton', 'it starts at the CNE and ends at the cenotaph in downtown toronto', 'it starts in downtown toronto and ends in ottawa', 'it starts in the city of Toronto and ends in the city of Hamilton', 'it starts in toronto and ends in ottawa', 'it starts in Toronto and ends in Ottawa']
**Perplexity**: 0.657 ✓
**LN-Entropy:** 0.260 ✓
**LexicalSimilarity:** 0.594 ✗
**SentBERTScore:** 0.149 ✗
**EigenScore:** -1.05 ✓
**EigenValue:** [3.31983018e+00 3.98560810e-01 2.17094299e-01 2.06965709e-01 1.53575354e-01 1.27925588e-01 7.82365136e-02 3.28158137e-02 1.01995086e-02 1.00000000e-03]

**Question:** which body part(s) occupy the greatest portion of the primary motor cortex
**GTAns:** face
**LLMAns:** the hand
**BatchGenerations:** ['the hand', 'the hand', 'the fingers', 'the hand', 'the hand', 'the hand and the foot', 'the hand', 'the hand and the foot', 'the hand and face', 'arms and legs']
**Perplexity**: 1.41 ✓
**LN-Entropy:** 0.300 ✓
**LexicalSimilarity:** 0.568 ✗
**SentBERTScore:** 0.163 ✗
**EigenScore:** -1.69 ✓
**EigenValue:** [3.76273036e+00 6.16284067e-01 1.96541049e-01 1.73505005e-01 1.28407153e-01 1.00000000e-03 1.00000000e-03 1.00000000e-03 1.00000000e-03 1.00000000e-03]

---

**Question:** who said have you no sense of decency
**GTAns:** Joseph Nye Welch
**LLMAns:** `Joseph Nye Welch`
**BatchGenerations:** ['Joseph N. Welch', 'Joseph N. Welch', 'joe stalin', 'joseph mccarthy', 'Joseph N. Welch', 'Joseph N. Welch', 'Joseph Nye Welch', 'joseph mccarthy', 'joe mccarthy', 'joseph mccarthy']
**Perplexity**: 0.666 ✗
**LN-Entropy:** 0.212 ✗
**LexicalSimilarity:** 0.437 ✗
**SentBERTScore:** 0.391 ✗
**EigenScore:** -1.85 ✓
**EigenValue:** [3.63114083e+00 8.11672323e-01 2.00385898e-01 3.19140618e-02 1.74251264e-02 1.00000000e-03 1.00000000e-03 1.00000000e-03 1.00000000e-03 1.00000000e-03]

---

## H.2 MODEL GENERATIONS WITH MANY AND FEW OUTLIERS

To demonstrate the relationship between the number of extreme features and model outputs, we provide several examples with many/few extreme features. The results show that when there are many extreme features, the model tends to generate consistent hallucination outputs across multiple generations. Instead, when there are few extreme features, the model generates diverse hallucination outputs which can be spotted by different hallucination detection metrics.

---

**Question:** who sang on great gig in the sky
**GTAns:** Clare Torry
**LLMAns:** `freddie mercury`
**AvgNumOutliers:** 15
**BatchGenerations:** ['freddie mercury', 'freddie mercury', 'freddie mercury', 'freddie mercury', 'freddie mercury', 'freddie mercury', 'freddie mercury', 'freddie mercury', 'freddie mercury', 'freddie mercury']
**Perplexity**: 0.263 ✗
**LN-Entropy:** 0.028 ✗
**LexicalSimilarity:** 1.0 ✗
**SentBERTScore:** 0.0 ✗
**EigenScore:** -2.63 ✗
**EigenValue:** [4.65740187e+00 1.00000000e-03 1.00000000e-03 1.00000000e-03 1.00000000e-03 1.00000000e-03 1.00000000e-03 1.00000000e-03 1.00000000e-03 1.00000000e-03]

---

**Question:** what are the top five wine producing states
**GTAns:** Washington
**LLMAns:** `California`
**AvgNumOutliers:** 13
**BatchGenerations:** [' California,', ' California,', ' california,', ' California,', ' California,', ' California,', ' California,', ' california,', ' California,', ' California,']
**Perplexity**: 0.368 ✗
**LN-Entropy:** 0.075 ✗
**LexicalSimilarity:** 1.0 ✗
**SentBERTScore:** 0.054 ✗
**EigenScore:** -2.42 ✗
**EigenValue:** [5.30709315e+00 1.13222379e-01 1.00000000e-03 1.00000000e-03 1.00000000e-03 1.00000000e-03 1.00000000e-03 1.00000000e-03 1.00000000e-03 1.00000000e-03]

---

---

**Question:** how many seasons of rules of engagement is there
**GTAns:** 7
**LLMAns:** 4 seasons
**AvgNumOutliers:** 2
**BatchGenerations:** ['3 seasons', '4 seasons', '4 seasons', '6 seasons', '7 seasons', '3 (2007-2009)', '3 (2007-2009)', '4 seasons', 'three', '11 seasons']
**Perplexity:** 0.996 ✓
**LN-Entropy:** 0.292 ✓
**LexicalSimilarity:** 0.307 ✓
**SentBERTScore:** 0.285 ✓
**EigenScore:** -1.60 ✓
**EigenValue:** [3.58548815e+00 5.87838054e-01 2.28057934e-01 1.36461300e-01 3.49712302e-02 1.11346059e-02 3.82259086e-03 1.00000000e-03 1.00000000e-03 1.00000000e-03]

---

**Question:** where did the first persian gulf war take place
**GTAns:** Israel
**LLMAns:** kuwait
**AvgNumOutliers:** 3
**BatchGenerations:** ['Iraq', 'Iraq and Kuwait', 'Iraq', 'kuwait', 'kuwait', 'in the middle east', 'in iraq', 'kuwait', 'iraq', 'kuwait']
**Perplexity:** 0.546 ✓
**LN-Entropy:** 0.281 ✓
**LexicalSimilarity:** 0.339 ✓
**SentBERTScore:** 0.224 ✓
**EigenScore:** -1.62 ✓
**EigenValue:** [3.59463352e+00 4.23782982e-01 2.57087067e-01 1.41513403e-01 6.20790226e-02 1.75980481e-02 1.00000000e-03 1.00000000e-03 1.00000000e-03 1.00000000e-03]

---

## H.3   IMPACT OF FEATURE CLIPPING

The texts in yellow represents model generations after applying feature clipping. The results show that after feature clipping, the overconfident generations can be appropriately suppressed, and some self-consistent hallucinations are finally identified.

---

**Question:** what are the top five wine producing states
**GTAns:** Washington
**LLMAns:** California
**BatchGenerations:** [' California,', ' California,', ' california,', ' California,', ' California,', ' California,', ' California,', ' california,', ' California,', ' California,']
**Perplexity:** 0.368 ✗
**LN-Entropy:** 0.075 ✗
**LexicalSimilarity:** 1.0 ✗
**SentBERTScore:** 0.054 ✗
**EigenScore:** -2.42 ✗
**EigenValue:** [5.30709315e+00 1.13222379e-01 1.00000000e-03 1.00000000e-03 1.00000000e-03 1.00000000e-03 1.00000000e-03 1.00000000e-03 1.00000000e-03 1.00000000e-03]
**BatchGenerations:** ['california', 'california', 'Washington', 'california', 'new york', 'california', 'washington', 'california', 'new york', 'michigan']
**EigenScore:** -1.32 ✓
**EigenValue:** [3.23392755e+00 8.41049340e-01 2.52322804e-01 1.33473529e-01 7.19449437e-02 6.12184197e-02 1.02734249e-02 5.33703500e-03 3.09878029e-03 1.00000000e-03]

**Question:** who sang on great gig in the sky
**GTAns:** Clare Torry
**LLMAns:** freddie mercury
**AvgNumOutliers:** 15
**BatchGenerations:** ['freddie mercury', 'freddie mercury', 'freddie mercury', 'freddie mercury', 'freddie mercury', 'freddie mercury', 'freddie mercury', 'freddie mercury', 'freddie mercury', 'freddie mercury']
**Perplexity**: 0.263 ✗
**LN-Entropy:** 0.028 ✗
**LexicalSimilarity:** 1.0 ✗
**SentBERTScore:** 0.0 ✗
**EigenScore:** -2.63 ✗
**EigenValue:** [4.65740187e+00 1.00000000e-03 1.00000000e-03 1.00000000e-03 1.00000000e-03 1.00000000e-03 1.00000000e-03 1.00000000e-03 1.00000000e-03 1.00000000e-03]
**BatchGenerations:** ['claire torry', 'freddie mercury', 'freddie mercury', 'freddie mercury', 'freddie mercury', 'freddie mercury', 'freddie mercury', 'freddie mercury', 'freddie mercury', 'freddie mercury']
**EigenScore:** -2.38 ✗
**EigenValue:** [4.38745800e+00 3.14982649e-01 1.00000000e-03 1.00000000e-03 1.00000000e-03 1.00000000e-03 1.00000000e-03 1.00000000e-03 1.00000000e-03 1.00000000e-03]

---

**Question:** who are you in assassin's creed 4
**GTAns:** third-person perspective
**LLMAns:** Edward Kenway
**BatchGenerations:** ['Edward Kenway', 'Edward Kenway', 'Edward Kenway', 'Edward Kenway', 'Edward Kenway', 'Edward Kenway', 'Edward Kenway', 'Edward Kenway', 'Edward Kenway', 'Edward Kenway']
**Perplexity**: 0.264 ✗
**LN-Entropy:** 0.002 ✗
**LexicalSimilarity:** 1.0 ✗
**SentBERTScore:** 0.0 ✗
**EigenScore:** -2.67 ✗
**EigenValue:** [2.10973201e+00 1.00000000e-03 1.00000000e-03 1.00000000e-03 1.00000000e-03 1.00000000e-03 1.00000000e-03 1.00000000e-03 1.00000000e-03 1.00000000e-03]
**BatchGenerations:** ['Edward Kenway', 'Edward Kenway', 'Connor', 'Edward Kenway', 'connor', 'Connor', 'alexander hamilton', 'Edward Kenway', 'ezio', 'connor']
**EigenScore:** -1.68 ✓
**EigenValue:** [3.47825477e+00 7.48127381e-01 3.24792650e-01 2.17182636e-01 8.15050807e-02 1.00000000e-03 1.00000000e-03 1.00000000e-03 1.00000000e-03 1.00000000e-03]

