# OpenReview forum: "INSIDE: LLMs' Internal States Retain the Power of Hallucination Detection"
_ICLR.cc/2024/Conference — ICLR 2024 poster_

### Official Review · Reviewer_5aJw · 2023-10-27

**Soundness:** 4 excellent
**Presentation:** 4 excellent
**Contribution:** 3 good
**Rating:** 8
**Confidence:** 4

**Summary:**

This paper proposed a new method for detection of LLM hallucinations. The method called EigenScore is based on extracting internal representation (layer activations) of the proposed sequences. Then, they compute the covariance matrix of K proposed sequences and then compute the logarithm determinant of the matrix. Intuitively, it measures entropy of the proposed sequences. They also propose the technique for reducing the number of confident hallucinations by clipping the outlier layer activations. In the experiments they empirically show that their proposed metric correlates with the correctness of the outputs more than other baselines such as Perplexity, Lexical similarity, Length normalized entropy and Energy.

**Strengths:**

The proposed method is simple, intuitive and it is well explained. The idea of looking into the internal states for hallucination detection is well motivated. I enjoyed the clear presentation of the paper. For example, the background section was quite useful for the understanding of the methods. The results seem to clearly show the advantage of the proposed method over several established baselines. The experiments are conducted on several diverse datasets, and several language models. The advantage of an alternative method on a benchmark with short answers is well explained and it is useful to inform the reader in which situations the proposed method should be employed. The ablation studies are quite complete and they show that the proposed method works under a wide range of settings.

**Weaknesses:**

Minor points and suggestions:
- Regarding the outliers in the activations, it would be nice to see if these outliers really lead to overconfident predictions. For example, is there any correlation between the number of outliers and accuracy of the answer? At least, it could be informative to see a few generated examples corresponding to many outliers or no outliers.
- For Table 2, it would be interesting to see if the quality of answers is increased after using clipping (as confident mistakes are excluded).
- In general, it would be nice to see some qualitative examples of hallucinations that are detected by the proposed method, but not detected by the baselines.
- centralized matrix -> centering matrix
- Typos: "prior works that rectifying", "nli-roterta-large".

**Questions:**

I have only minor questions and suggestions and I would like to hear authors' discussion on those points.

---

> ### Author Response · Authors · 2023-11-19
> **Response to Reviewer 5aJw (R4)**
>
> **Comment** We thank Reviewer 5aJw (R4) for their constructive suggestions. According to the suggestion, we have provided a comprehensive Case Studies in the **Appendix H (Page16-21)** of our revised paper. Here is a point-by-point response.
>
> ***
>
> > **W1** Regarding the outliers in the activations, it would be nice to see if these outliers really lead to overconfident predictions. For example, is there any correlation between the number of outliers and accuracy of the answer? At least, it could be informative to see a few generated examples corresponding to many outliers or no outliers.
>
> **Ans** Thanks for your suggestion. The outliers features increase the likelihood of generating overconfident and self-consistent generations. For example, the top 50 tokens that contain the most outliers have an average confidence of 0.98, while the bottom 50 tokens that contain the least outliers have an average confidence of 0.54. According to your suggestion, we provide several examples that contain many/few extreme features in the **Appendix H.2 (Page19)**. The results show that when there are many extreme features, the model tends to generate consistency hallucination outputs across multiple generations. Instead, when there are few extreme features, the model generates diverse hallucination outputs which can be spotted by different hallucination detection metrics.
>
> ***
>
> > **W2** For Table 2, it would be interesting to see if the quality of answers is increased after using clipping (as confident mistakes are excluded).
>
> **Ans** It is worth noting that the feature clipping approach is proposed to prevent the model from generating overconfident outputs for uncertain questions, but not to improve the quality of answers. To demonstrate the effectiveness of the feature clipping strategy, we show several  cases before and after feature clipping in the **Appendix H.3 (Page20)**
>
> ***
>
> > **W3** In general, it would be nice to see some qualitative examples of hallucinations that are detected by the proposed method, but not detected by the baselines.
>
> **Ans** Thanks for your suggestion. We have provided comprehensive cases in **Appendix H.1 (Page16)**, including good cases and failure cases. Please see the details in our revised paper.
>
> ***
>
> > **W4** typos：centralized matrix -> centering matrix；"prior works that rectifying", "nli-roterta-large".
>
> **Ans** Thanks，the typos have been revised. We will re-polish our paper carefully.

---

### Official Review · Reviewer_9Srq · 2023-10-30

**Soundness:** 2 fair
**Presentation:** 3 good
**Contribution:** 3 good
**Rating:** 6
**Confidence:** 4

**Summary:**

This work follows the line of uncertainty-based methods for hallucination detection to probe the inconsistency between the independently generated response for the same query. The idea is to take the eigenvalues of LLM internal states' covariance matrix to measure the inconsistency.

**Strengths:**

If we don't consider concurrent works, such as "Representation Engineering: A Top-Down Approach to AI Transparency", the proposed method is novel and feasible since measuring covariance between several independent generations is intuitive.

The proposed method is simple and easy to implement and use without tuning models, and the performance improvement is significant.

**Weaknesses:**

The experiment setting could be improved since this work targets hallucination detection, which is a broad area, and there are various types of datasets. Only reporting results on QA benchmarks may be insufficient to support the contribution. At least, I think truthfulQA is closer to the topic of hallucination compared to the benchmarks used in the paper. Additionally, it's better to include recent LLMs, such as LLaMA2 and Falcon.

The covariance of LLM internal states may contain distracting information, such as the diversity of expressions, styles, and terminology. Since there are multiple factors that influence the difference between internal states, it's hard to guarantee the generalization ability of the proposed method.

**Questions:**

For the correctness Measure, I agree that the evaluation of answers is challenging, but it's better to have an exact match score so that we can compare it with other existing works.

As feature clipping is claimed as a contribution of this work, could you provide some ablation studies for it? I am curious why it needs a dynamic memory bank. Does that mean the distribution of activation values changes a lot across samples?

---

> ### Author Response · Authors · 2023-11-19
> **Response to Reviewer 9Srq (R3) -- Part1**
>
> **Comment** We thank Reviewer 9Srq (R3) for the careful reviews and insightful suggestions, which really helped us improve our paper.
>
> >**W1** (1) The experiment setting could be improved. I think TruthfulQA is closer to the topic of hallucination compared to the benchmarks used in the paper. (2) It's better to include recent LLMs, such as LLaMA2 and Falcon.
>
> **Ans** (1) According to your suggestion, we have compared our proposal with several competitve approaches in the TruthfulQA dataset. The results are presented in Table 1. For the ITI [1], which provides a strong hallucination detector that trained on TruthfulQA, we report the best performance in their paper. As can be seen, our proposal consistently outperforms the baseline methods and achieves comparable performance as ITI when we utilize 50 in-distribution prompts.
>
> Table 1: Performance comparison of different methods on TruthfulQA dataset. Accuracy is reported.
> | #prompt | Perplexity | LN-Entropy | LexialSim | SelfCKGPT | ITI* | **EigenScore** |
> |:-------:|:----------:|:----------:|:---------:|:---------:|:---------:|:--------------:|
> |    5    |    70.0    |    71.2    |    73.6   |    74.2   |  **83.3** |      76.7      |
> |    20   |    76.4    |    77.7    |    77.9   |    76.8   |  **83.3** |      79.5      |
> |    50   |    73.1    |    77.9    |    73.6   |   78.3    |  **83.3** |    **81.3**    |
>
> It's worth nothing that the ITI [1] relies on training 1024 binary classifiers in TruthQA datasets, and they report the best performance (83.3) in the validation set. Therefore, their best performance is better than our method which has not been trained on TruthfulQA. However, training on the validation set also limits the generalization of their method on other domains [1].
>
> As TruthfulQA is a very challenging dataset for LLMs, zero-shot inference results in poor performance. Therefore, we follow previous work [2] to utilize different number of in-distribution prompts during inference time. The results show that the performance could be significantly improved when we increase the number of in-distribution prompts, which also explains why ITI performs good.
>
> Please see the **Appendix A (Page13)** of our revised paper for the detailed results and analysis.
>
> (2) In the main experiments, we evaluate the performance of different methods in LLaMA-7B, LLaMA-13B and OPT-6.7B. To demonstrate the robustness of our method across different models, we also provide the performance comparison in the recent LLaMA2-7B  and Falcon-7B. The results in Table 2 reveal that our proposal consistently exhibits superior performance compared to the competitive methods across different LLMs.
>
> Table2: Performance evaluation on LLaMA2-7B and Falcon-7B. AUC$_s$/AUC$_r$ score is reported.
> |  Methods  | Datasets | Perplexity | LN-Entropy | Lexical Similarity | SelfCheckGPT |   EigenScore  |
> |:---------:|----------|:----------:|:----------:|:------------------:|:------------:|:-------------:|
> | LLaMa2-7b |   CoQA   |  62.2/66.6 |  69.9/75.2 |      74.4/77.5     |   72.4/75.1  | **78.6/80.7** |
> | LLaMa2-7b |    NQ    |  70.8/70.2 |  72.1/71.2 |      72.1/72.9     |   69.1/68.1  | **74.4/73.7** |
> | Falcon-7b |   CoQA   |  57.0/60.6 |  62.6/63.2 |      74.8/76.4     |   76.7/77.9  | **80.8/80.6** |
> | Falcon-7b |    NQ    |  74.3/74.7 |  74.6/74.7 |      73.8/75.4     |   74.7/74.0  | **76.3/75.7** |
>
> For more information, please see the **Appendix C (Page13)** of our revised paper.
>
> ***
>
> > **W2** The covariance of LLM internal states may contain distracting information (such as the diversity of expressions, styles, and terminology), it's hard to guarantee the generalization ability of the proposed method.
>
> **Ans**  The diversity of natural language expressions has long presented a challenge in evaluating neural language generation and detecting hallucinations. Our proposal, of course, cannot completely solve the problem. However, compared to existing uncertainty or consistency metrics, our proposal takes a step further in solving this problem: (1) It captures the semantic divergence (entropy) in the dense embedding space, which is expected to retain highly-concentrated semantic information.  (2) The proposed EigenScore, which is determined by the LogDet of the sentence embeddings' Covariance, is more intuitive and effective compared to existing metrics that explores sentence-wise similarity
>
> Besides, As presented in our experimental setup, we set the temperature t=0.5 through the experiments, which appropriately suppressed the diversity of LLMs' generations. In Fig. 4(a) (of our submission), we have presented the performance sensitivity to temperature t, which shows that our EigenScore significantly outperforms baseline methods when t<1， and when t>1 the performance decreases rapidly.
>
> ***
> **Reference**
>
> [1] TruthfulQA: Measuring How Models Mimic Human Falsehoods
>
> [2] Training a helpful and harmless assistant with reinforcement learning from human feedback

---

> ### Author Response · Authors · 2023-11-19
> **Response to Reviewer 9Srq (R3) -- Part2**
>
> > **Q1** For the correctness Measure, I agree that the evaluation of answers is challenging, but it's better to have an exact match score so that we can compare it with other existing works
>
> **Ans** According to the suggestion, we have provided the evaluation results by employing Exact Match as the correctness measure. The results in Table 3 show similar conclusion with using ROUGE and sentence similarity as the correctness measure, which demonstrates that our proposal consistently outperforms the comparison baselines under different evaluation metrics.
>
> Please see the **Appendix E (Page14)** in our revised paper for more details.
>
> Table3: Performance evaluation with Exact Match as correctness measure.
> |  Models  |  Methods | Perplexity | LN-Entropy | Lexical Similarity | EigenScore |
> |:--------:|:--------:|:----------:|:----------:|:------------------:|:----------:|
> | LLaMA-7B |   CoQA   |    63.7    |    70.7    |        76.1        |  **83.0**  |
> | LLaMA-7B |   SQuAD  |    57.3    |    72.1    |        76.9        |  **83.9**  |
> | LLaMA-7B |    NQ    |    75.3    |    75.6    |        75.8        |  **80.1**  |
> | LLaMA-7B | TriviaQA |    82.5    |  **83.4**  |        81.8        |    82.4    |
> | OPT-6.7B |   CoQA   |    59.4    |    61.7    |        71.8        |  **79.4**  |
> | OPT-6.7B |   SQuAD  |    56.7    |    65.2    |        72.7        |  **82.9**  |
> |OPT-6.7B |    NQ    |  **79.8**  |    78.1    |        73.2        |  **79.8**  |
> | OPT-6.7B | TriviaQA |  **83.8**  |    81.3    |        79.3        |    82.7    |
>
> ***
>
> > **Q2** (1) As feature clipping is claimed as a contribution of this work, could you provide some ablation studies for it? (2) I am curious why it needs a dynamic memory bank. Does that mean the distribution of activation values changes a lot across samples?
>
> **Ans** (1) We have already provided the ablation study in our original submission in (**Page7 - Effectiveness of Feature Clipping**), which demonstrates the effectiveness of the introduced feature clipping techinque. Besides, we also add several samples to show the hallucination detection results before and after feature clipping in the **Appendix H3 (page18)**.
>
> (2) We illustrate the distributions of neuron activation from four selected tokens in **Fig.6 (Appendix F, Page15)**, which shows that the distribution changes a lot across samples. Therefore, it is risky to determine the clipping threshold with only the current input sample (**EigenScore-C**). A feasible solution is to pre-compute the optimal threshold based on a batch of input samples (**EigenScore-P**). Besides, another solution is to dynamically record the activation values and determine the threshold during the inference process (**EigenScore-MB**). We have tried both solutions and the experimental results are presented in Table 4. The results show that determining the thresholds with a memory bank works slightly better. We attribute this variability to potential differences in the activation distributions across various datasets.
>
> Table4: Ablation study of determining the clipping threshold with different technique. AUC$_s$ is reported.
> |      Methods     |   CoQA   |    NQ    |
> |:----------------:|:--------:|:--------:|
> |   EigenScore-C |   78.1   |   74.8   |
> |   EigenScore-P   |   79.9   |   75.3   |
> |   EigenScore-MB  | **80.4** | **76.5** |
>
> Please see the **Appendix F (Page15)** in our revised paper for more details.
>
> ***
>
> **Reference**
>
> [1] TruthfulQA: Measuring How Models Mimic Human Falsehoods
>
> [2] Training a helpful and harmless assistant with reinforcement learning from human feedback

---

### Official Review · Reviewer_Kmk8 · 2023-11-03

**Soundness:** 3 good
**Presentation:** 3 good
**Contribution:** 3 good
**Rating:** 6
**Confidence:** 4

**Summary:**

This paper proposes a new framework called INSIDE that leverages the internal states of large language models (LLMs) to detect hallucinated content generated by the LLM. A key component is the EigenScore metric, which measures semantic consistency in the embedding space and represents differential entropy. This helps identify overconfident and contradictory generations. They also introduce a test time feature clipping approach that truncates extreme activations, thereby reducing overconfidence and enabling better hallucination detection. Experiments on several QA benchmarks show state-of-the-art performance in hallucination detection. Ablation studies verify the efficacy of the proposed techniques, including the EigenScore metric and feature clipping. Overall, INSIDE provides an effective way to detect hallucinated content from LLMs by leveraging their internal states and controlling overconfidence. The EigenScore and feature clipping are two key innovations that improve hallucination detection.

**Strengths:**

1. The paper is very interesting and easy to follow
2. The paper poses a unique angle to detect hallucination from internal states with eigenscore
3. The idea is overall innovative while some key points are inspired by previous work
4. The experiments are solid to support the paper's claim

**Weaknesses:**

1. One of the obvious shortcomings of the proposed method is the applicability to blackbox LLMs, e.g., GPT, PaLM and Claude.  The method relies on the internal state of the model, e.g., LLaMA and OPT, but the authors may not discuss the limitation well.

2. The computational cost is another concern since essentially this method is a sampling-based approach. While the number of samples is not large, how to further reduce the cost for real production is still unsolved.

3. The proposed method is mainly inspired by uncertainty estimation but how to bridge the connection with hallucination detection is still unclear, even though the results look promising.

4. Beyond hallucination detection, is that possible to mitigate hallucination through the proposed framework eventually?

**Questions:**

1. How about the proposed method compared with this work [1] on the TruthfulQA dataset?  Both work relies on the internal state of the LLM models.

[1] Li, Kenneth, Oam Patel, Fernanda Viégas, Hanspeter Pfister, and Martin Wattenberg. "Inference-Time Intervention: Eliciting Truthful Answers from a Language Model." arXiv preprint arXiv:2306.03341 (2023).

2. How about the effect of sentence embeddings on the final performance?

3. Can you provide the computational cost estimation of the whole framework and which type of resources you are using?

4. Since the evaluation metric is AUROC, I am wondering if, can you compare the proposed method with the GPT 3.5 or 4 performance. Or can you add some comparison with the hallucination detection baseline methods, e.g., selfcheckGPT as you mentioned in the paper?

5. The current baselines are mainly from the authors' choice without open baselines, e.g., semantic uncertainty (ICLR 2023). I noted that many related works are good baselines that can be used for comparison.


---
post rebuttal

I think the authors' response and experiments addressed my concerns so I raise my evaluation to support my decision.

---

> ### Author Response · Authors · 2023-11-19
> **Response to Reviewer Kmk8 (R2) -- Part1**
>
> **Comment** We thank Reviewer Kmk8 (R2) for the careful reviews and insightful suggestions, which helped us improve our paper.
>
> > **W1** One shortcoming of the proposed method is the applicability to blackbox LLMs. The method relies on the internal state of the model, e.g., LLaMA and OPT, but the authors may not discuss the limitation well.
>
> **Ans** The reliance on white-box LLMs is a limitation of our work, and thanks for your suggestion, we have discussed the limitations of our work in the revised paper (**Appendix G, Page 16**).
> In addition, black-box based [1,8] and white-box based [2,3] methods are equally important for different application scenarios for LLMs. Compared to the methods designed for black-box methods, exploring the internal information for hallucination detection has its own advantages: (1) The internal states retain highly-concentrated semantic information; (2) Exploring the sentence embedding in the internal states is much more efficient compared to extracting the sentence embedding with an additional large model [2,3].
>
> ***
>
> > **W2** The computational cost is another concern since essentially this method is a sampling-based approach. How to further reduce the cost for real production is still unsolved.
>
> **Ans** One limitation of our work is that it requires additional computational overhead，which is also the limitation for all consistency-based [2] or prompting-based methods [7]. We have discussed this limitation in our revised paper (**Appendix G, Page16**).
>
> Besides, we also compare and analysis the computational overhead of our proposal with the base LLM and other compared baselines. The average inference time (second per question) is computed on one NVIDIA-A100 GPU and shown in Table 1. As observed, our EigenScore is about 10 times more efficient than the methods that rely on another large model to measure the self-consistency (such as SelfCheckGPT [2]), and shares the similar computational overhead with the LN-Entropy and Lexical Similarity. It is worth noting that the inference overhead required to generate multiple results is not linearly proportional to the time of generating a single output, owing to the sampling and decoding strategy of the autoregressive LLM model.
>
> Table 1: Inference cost comparison of different methods in LLaMA-7B and LLaMA-13B.
> |   Methods | BaseLLM | Perplexity | LN-Entropy | Lexical Similarity | SelfCheckGPT | **EigenScore** |
> |:---------:|:-------:|:----------:|:----------:|:------------------:|:------------:|:--------------:|
> |  LLaMA-7B |  0.245s |   0.245s   |   0.803s   |       0.815s       |    10.68s    |     0.805s     |
> | LLaMA-13B |  0.309s |   0.307s   |    1.27s   |        1.28s       |    10.26s    |      1.27s     |
>
>
> Compared to the computational overhead of generating multiple outputs, the cost of feature clipping and EigenScore computation is negligible (0.06s). For more details about the **Computational Efficiency Analysis**, please see the **Appendix D (Page14)** of our revised paper.
>
> ***
> >**W3** The proposed method is mainly inspired by uncertainty estimation but how to bridge the connection with hallucination detection is still unclear, even though the results look promising.
>
> **Ans**  LLMs' knowledge hallucinations primarily arise from uncertainties regarding the source information or uncertainties inherent in the decoding process [5]. Therefore, exploring the uncertainty [5,6] or self-consistency [2,7] of LLMs' generations is one of the most promising direction for effective hallucination detection. According to [7], the self-contradictory hallucinations is one of the most important form of hallucination, and **67%** inconsistent generation indicates hallucination. Therefore, we suggest that our proposal exploring semantic consistency with LLMs' internal information is potentially a promising approach for hallucination detection.
>
> ***
>
> **Reference**
>
> [1] Inference-Time Intervention: Eliciting Truthful Answers from a Language Model.
>
> [2] SelfCheckGPT: Zero-resource black-box hallucination detection for generative large language models.
>
> [3] Semantic Uncertainty: Linguistic Invariances for Uncertainty Estimation in Natural Language Generation.
>
> [4] Shifting attention to relevance: Towards the uncertainty estimation of large language models.
>
> [5] Survey of Hallucination in Natural Language Generation.
>
> [6] On Hallucination and Predictive Uncertainty in Conditional Language Generation.
>
> [7] Self-contradictory Hallucinations of Large Language Models: Evaluation, Detection and Mitigation.
>
> [8] Representation engineering: A top-down approach to ai transparency.
>
> [9] Insights into Classifying and Mitigating LLMs' Hallucinations
>
> [10] Training a helpful and harmless assistant with reinforcement learning from human feedback.

---

> ### Author Response · Authors · 2023-11-19
> **Response to Reviewer Kmk8 (R2) -- Part2**
>
> >**W4** Beyond hallucination detection, is that possible to mitigate hallucination through the proposed framework eventually?
>
> **Ans** In most studies, the hallucination detection and mitigatition are two independent modules [3]. Most hallucination mitigatition methods can be categorized into prompting-based, external knowledge based and training/model-editing based methods [9]. In this study, we mainly focus on the hallucination detection. However, the hallucination content can be appropriately suppressed by the well-established prompting-based or external knowledge based approaches after identifying the hallucination content with EigenScore. For exapmle, in [7], the authors use a simple prompt template to correct the detected self-contradiction hallucination.
>
> In future work, we will delve into how to effectively mitigate hallucinations with our EigenScore.
>
> ***
>
> >**Q1** How about the proposed method compared with this work [1] on the TruthfulQA dataset? Both work relies on the internal state of the LLM models.
>
> **Ans** Considering that replicating the ITI [1] would be prohibitively expensive, we test the performance of our proposal and comparison baselines in the TruthfulQA dataset. The results are presented in Table 2. For the ITI, we report the best performance in their paper. As can be seen, our proposal consistently outperforms the baseline methods and achieves comparable performance as ITI when we utilize 50 in-distribution prompts.
>
> Table 2: Performance comparison of different methods on TruthfulQA dataset. Accuracy is reported.
> | #prompt | Perplexity | LN-Entropy | LexialSim | SelfCKGPT | ITI* | **EigenScore** |
> |:-------:|:----------:|:----------:|:---------:|:---------:|:---------:|:--------------:|
> |    5    |    70.0    |    71.2    |    73.6   |    74.2   |  **83.3** |      76.7      |
> |    20   |    76.4    |    77.7    |    77.9   |    76.8   |  **83.3** |      79.5      |
> |    50   |    73.1    |    77.9    |    73.6   |   78.3    |  **83.3** |    **81.3**    |
>
> It's worth nothing that the ITI [2] relies on training 1024 binary classifiers in TruthQA datasets. Therefore, their best performance is better than our method which has not been trained on TruthfulQA. However, training on the validation set also limits the generalization of their method on other domains [2].
>
> As TruthfulQA is a very challenging dataset for LLMs, zero-shot inference results in poor performance. Therefore, we follow previous work [10] to utilize different number of in-distribution prompts during inference time. The results show that the performance could be significantly improved when we increase the number of prompts, which also explains why ITI performs good.
>
> For more information about the experiments, please see the **Appendix A (Page13)** of our revised paper.
>
> ***
>
> >**Q2** How about the effect of sentence embeddings on the final performance?
>
> **Ans** We have performed this ablation study in **Page8 - How EigenScore Performs with Different Sentence Embeddings**. The ablation results presented in **Fig.3 (b)** shows that using the sentence embedding in the shallow and final layers yields significantly inferior performance compared to using sentence embedding in the layers close to the middle. Besides, another interesting observation is that utilizing the embedding of the last token as the sentence embedding achieves superior performance compared to simply averaging the token embeddings, which suggests that the last token of the middle layers retain more  information about the truthfulness of generations.
>
> ***
>
> >**Q3** Can you provide the computational cost estimation of the whole framework and which type of resources you are using?
>
> **Ans**  Please refer to our response to **W2**. We have provided a detailed **Computational Efficiency Analysis** in **Appendix D (Page14)** of our revised paper. All experiments are performed on the NVIDIA-A100 GPU.
>
> ***
>
> **Reference**
>
> [1] Inference-Time Intervention: Eliciting Truthful Answers from a Language Model.
>
> [2] SelfCheckGPT: Zero-resource black-box hallucination detection for generative large language models.
>
> [3] Semantic Uncertainty: Linguistic Invariances for Uncertainty Estimation in Natural Language Generation.
>
> [4] Shifting attention to relevance: Towards the uncertainty estimation of large language models.
>
> [5] Survey of Hallucination in Natural Language Generation.
>
> [6] On Hallucination and Predictive Uncertainty in Conditional Language Generation.
>
> [7] Self-contradictory Hallucinations of Large Language Models: Evaluation, Detection and Mitigation.
>
> [8] Representation engineering: A top-down approach to ai transparency.
>
> [9] Insights into Classifying and Mitigating LLMs' Hallucinations
>
> [10] Training a helpful and harmless assistant with reinforcement learning from human feedback.

---

> > ### Author Response · Authors · 2023-11-19
> > **Response to Reviewer Kmk8 (R2) -- Part3**
> >
> > > **Q4/5** (1) Can you add some comparison with the hallucination detection baseline methods, e.g., selfcheckGPT as you mentioned in the paper? (2) The current baselines are mainly from the authors' choice without open baselines, e.g., semantic uncertainty (ICLR 2023). I noted that many related works are good baselines that can be used for comparison.
> >
> > **Ans** To demonstrate the effectiveness of our proposal, we also compare our EigenScore with several competitive methods, including **Semantic Entropy (SemanticEnt)** [3], Shifting Attention to Relevance (**SentSAR**) [4] and SelfCheckGPT (**SelfCKGPT**) [2]. The comparison results in Table 3 demonstrate that our EigenScore significantly outperforms the competitor.
> >
> > Table 3: Performance comparison of EigenScore and and several competitive methods on CoQA dataset. AUC$_s$/AUC$_r$ score is reported.
> > |       Methods      | SemanticEn |  SentSAR  | SelfCkGPT |   EigenScore  |
> > |:------------------:|:----------:|:---------:|:---------:|:-------------:|
> > | OPT-6.7B |  63.1/71.7 | 69.8/72.2 | 70.2/74.1 | **71.9/77.5** |
> > |  LLaMA-7B |  64.9/68.2 | 70.4/65.8 | 68.7/72.9 | **71.2/75.7** |
> > | LLaMA-13B |  65.3/66.7 | 71.4/64.7 | 68.1/77.0 | **72.8/79.8** |
> >
> > Note that both SentSAR, SelfCheckGPT and our proposal evaluate the quality of LLMs' generation by exploring the semantic consistency across multiple outputs. In contrast to Semantic Entropy and SelfCheckGPT which rely on another language model for sentence embedding extraction, our approach leverages the internal states of LLMs, which retain highly-concentrated semantic information.
> >
> > Additionally, the EigenScore defined by the LogDet of the sentence covariance matrix is able to capture the semantic consistency more effectively compared to the sentence-wise similarity [2]. Furthermore, the proposed feature clipping strategy allows our model to identify the overconfident hallucinations, which has not been investigated by existing methods [2,3,4]
> >
> > Please see the **Appendix B (Page13)** for the details about the comparison results and analysis.
> >
> > ***
> > **Reference**
> >
> > [1] Inference-Time Intervention: Eliciting Truthful Answers from a Language Model.
> >
> > [2] SelfCheckGPT: Zero-resource black-box hallucination detection for generative large language models.
> >
> > [3] Semantic Uncertainty: Linguistic Invariances for Uncertainty Estimation in Natural Language Generation.
> >
> > [4] Shifting attention to relevance: Towards the uncertainty estimation of large language models.
> >
> > [5] Survey of Hallucination in Natural Language Generation.
> >
> > [6] On Hallucination and Predictive Uncertainty in Conditional Language Generation.
> >
> > [7] Self-contradictory Hallucinations of Large Language Models: Evaluation, Detection and Mitigation.
> >
> > [8] Representation engineering: A top-down approach to ai transparency.
> >
> > [9] Insights into Classifying and Mitigating LLMs' Hallucinations
> >
> > [10] Training a helpful and harmless assistant with reinforcement learning from human feedback.

---

> > > ### Comment · Reviewer_Kmk8 · 2023-11-23
> > > **Thanks for your detailed reply**
> > >
> > > Dear authors,
> > >
> > > Thanks for your detailed responses with additional experiments.  These efforts have addressed my concerns and questions so I decided to raise my score to 6, reflecting my support for this work.

---

### Official Review · Reviewer_VqqH · 2023-11-09

**Soundness:** 3 good
**Presentation:** 3 good
**Contribution:** 2 fair
**Rating:** 6
**Confidence:** 3

**Summary:**

This work proposes a metric, EigenScore, to measure whether the LLM can answer the question correctly. EigenScore is calculated from internal states of multiple responses to the same question. Furthermore, the authors design a feature clipping approach to truncate the extreme activations of internal states to reduce the overconfident generations. This benefits mitigating overconfident hallucination. The empirical results show the effectiveness of this work in detecting hallucinations.

**Strengths:**

1. This paper studies a timely and important research problem in llm. Preventing hallucinations in inference time is a huge research challenge in the community.
2. This paper is easy to follow.
3. The proposed approach is effect in detecting hallucinations.

**Weaknesses:**

1. The related work comparison is not comprehensive. There is no related work comparison between the internal-state detection approach. For example, [1] investigates mitigating the hallucinations by editing the internal states of llm. They first measure the possibility of generating hallucinations by observing the internal states and then editing those internal states. It might be better if the authors can compare their approach with the internal state-based hallucination detection approaches, e.g., [1].
2. The llm models involve multiple transformation blocks; which block internal state do the authors investigate? It would make the paper more impactful if the authors could present the ablation on transformation blocks.

[1] Li, Kenneth, et al. "Inference-Time Intervention: Eliciting Truthful Answers from a Language Model." NeurIPS 2023.

**Questions:**

1. I find that the evaluation metrics are related to fact-checking. I wonder if there are diverse metrics to evaluate the hallucinations.

---

> ### Author Response · Authors · 2023-11-19
> **Response to Reviewer VqqH (R1) -- Part1**
>
> **Comment** We thank the Reviewer VqqH (R1) for the insightful feedbacks.
>
> > **W1** The related work comparison is not comprehensive. There is no related work comparison between the internal-state detection approach. It might be better if the authors can compare their approach with the internal state-based hallucination detection approaches, e.g., [1].
>
> **Ans** Thanks for your suggestion. Considering that replicating the ITI [2] would be prohibitively expensive, we test the performance of our proposal and the baselines in the TruthfulQA dataset [1]. The results are presented in Table 1. For the ITI, we report the best performance in their paper. As can be seen, our proposal consistently outperforms the baseline methods and achieves comparable performance as ITI when we utilize 50 in-distribution prompts.
>
> It's worth nothing that the ITI [2] relies on training 1024 binary classifiers in TruthfulQA datasets, and they report the best performance (83.3) in the validation set. Therefore, their best performance is better than our method which has not been trained on TruthfulQA. However, training on the validation set also limits the generalization of their method on other domains [2].
>
> Table 1: Performance comparison of different methods on TruthfulQA dataset. Accuracy is reported.
> | #prompt | Perplexity | LN-Entropy | LexialSim | SelfCKGPT | ITI* | **EigenScore** |
> |:-------:|:----------:|:----------:|:---------:|:---------:|:---------:|:--------------:|
> |    5    |    70.0    |    71.2    |    73.6   |    74.2   |  **83.3** |      76.7      |
> |    20   |    76.4    |    77.7    |    77.9   |    76.8   |  **83.3** |      79.5      |
> |    50   |    73.1    |    77.9    |    73.6   |   78.3    |  **83.3** |    **81.3**    |
>
> As TruthfulQA is a very challenging dataset for LLMs, zero-shot inference results in poor performance. Therefore, we follow previous work [6] to utilize different number of in-distribution prompts during inference time. The results show that the performance could be significantly improved when we increase the number of prompts, which also explains why ITI performs good.
>
> Besides, to demonstrate the effectiveness of our proposal, we also compare our proposal with several competitive methods [3,4,5] that explore self-consistency for hallucination detection. The results in Table 2 demonstrate that our EigenScore consistently outperforms the the competitors by a large margin.
>
> Table 2: Performance comparison of EigenScore and and several competitive methods on CoQA dataset. AUC$_s$/AUC$_r$ score is reported.
> |       Methods      | SemanticEn |  SentSAR  | SelfCkGPT |   EigenScore  |
> |:------------------:|:----------:|:---------:|:---------:|:-------------:|
> | OPT-6.7B  |  63.1/71.7 | 69.8/72.2 | 70.2/74.1 | **71.9/77.5** |
> |  LLaMA-7B |  64.9/68.2 | 70.4/65.8 | 68.7/72.9 | **71.2/75.7** |
> | LLaMA-13B |  65.3/66.7 | 71.4/64.7 | 68.1/77.0 | **72.8/79.8** |
>
> For more information about the experiments, please see the **Appendix A/B (Page13)** of our revised paper.
>
> ***
>
> > **W2** The llm models involve multiple transformation blocks; which block internal state do the authors investigate? It would make the paper more impactful if the authors could present the ablation on transformation blocks.
>
> **Ans** We have demonstrated the performance of exporing EigenScore in different layers (transformer blocks) in **Fig. 3(b)**. And the detailed discussions are illustrated on **Page8** in our submission. The results show that using the sentence embedding in the shallow and final layers yields significantly inferior performance compared to using sentence embedding in the layers close to the middle. Besides, another interesting observation is that utilizing the embedding of the last token as the sentence embedding achieves superior performance compared to simply averaging the token embeddings, which suggests that the last token of the middle layers retain more information about the truthfulness of generations.
>
> ***
> **Reference**
>
> [1] TruthfulQA: Measuring How Models Mimic Human Falsehoods
>
> [2] Inference-Time Intervention: Eliciting Truthful Answers from a Language Model
>
> [3] Selfcheckgpt: Zero-resource black-box hallucination detection for generative large language models
>
> [4] Semantic Uncertainty: Linguistic Invariances for Uncertainty Estimation in Natural Language Generation
>
> [5] Shifting Attention to Relevance: Towards the Uncertainty Estimation of Large Language Models
>
> [6] Training a helpful and harmless assistant with reinforcement learning from human feedback

---

> > ### Author Response · Authors · 2023-11-19
> > **Response to Reviewer VqqH (R1) -- Part2**
> >
> > > **Q1** I find that the evaluation metrics are related to fact-checking. I wonder if there are diverse metrics to evaluate the hallucinations.
> >
> > **Ans** Both hallucination detection and fact-checking aim at evaluating the quality of LLMs' generations. Consequently, the evaluation metrics for factual evaluation and hallucination detection tasks are similar. We following exiting works [3,4,5] to utlize the AUC and Pearson Correlation Coefficient to evaluate the hallucination detection performance. Besides, for the experiments in TruthfulQA dataset, we also introduce accuracy as an evaluation metric.
> >
> > ***
> > **Reference**
> >
> > [1] TruthfulQA: Measuring How Models Mimic Human Falsehoods
> >
> > [2] Inference-Time Intervention: Eliciting Truthful Answers from a Language Model
> >
> > [3] Selfcheckgpt: Zero-resource black-box hallucination detection for generative large language models
> >
> > [4] Semantic Uncertainty: Linguistic Invariances for Uncertainty Estimation in Natural Language Generation
> >
> > [5] Shifting Attention to Relevance: Towards the Uncertainty Estimation of Large Language Models
> >
> > [6] Training a helpful and harmless assistant with reinforcement learning from human feedback

---

### Author Response · Authors · 2023-11-19
**Gloabl Response**

**Comment** We sincerely thank all the reviewers for their careful reviews and constructive suggestions, which helped us improve our submission. Here, we provide a comprehensive overview of the reviewers' feedback and outline the corresponding modifications we have made.

## The merits of our submission

The reviewers acknowledge the strengths of our paper as follows:
* The method is intuitive, innovative, easy-to-follow, and well-explained.
* The contribution and presentation are good/excellent
* The experiments are solid to support the paper's claim
* The performance improvement is significant.


## Important suggestions and concerns

Some important suggestions and concerns are:
* Performance evaluation on other benchmark (TruthfulQA)
* Introducing more comparison methods
* Evaluation on more recent LLMs.
* Applicability to blackbox and inference cost
* More correctness evaluation measure (Exact Match)
* Add some case study


## Important information missed by reviewers in our original submission

Note that, some reviewers might have overlooked certain crucial ablation experiments in the original submission. In our first submission, we have provided the ablation on **How EigenScore Performs with Different Sentence Embeddings** in **Page8** and the **Effectiveness of Feature Clipping** in **Page7**.


## Our Revisions

According to the reviewers' constructive and insightful feedbacks, we have carefully revised our paper, and provided comprehensive experiments, analysis and case analysis in the appendix.

* In the **Appendix A (Page13)**, we compare our proposal and several baseline methods in TruthfulQA dataset.
* In the **Appendix B (Page13)**, we compare our EigenScore with several competitive approaches.
* In the **Appendix C (Page14)**, we evauate our method in the recent LLaMA2 and Falcon models.
* In the **Appendix D (Page14)**, we compare the inference cost of different methods and provide a computation efficiency analysis.
* In the **Appendix E (Page15)**, we perform more results evaluating the correctness by Exact Match
* In the **Appendix F (Page15)**, we provide more visualization and ablation for Feature Clipping.
* In the **Appendix G (Page16)**, we discuss the limitations of our work
* In the **Appendix H (Page16-21)**, we provide comprehensive case studies


Finally, I sincerely thank all reviewers and AC for their efforts.

---

> ### Author Response · Authors · 2023-11-21
> **A Gentle Reminder**
>
> Dear Reviewers:
>
> Thank you for your valuable and constructive feedbacks, which have inspired further improvements to our paper. As a gentle reminder, it has been more than 2 days since we submitted our detailed responses and revised paper. As the rebuttal deadline is approaching, we would greatly appreciate knowing if our response adequately addressed your concerns. We are looking forward to your response and are happy to answer any future questions.
>
>
> Best,
>
> Authors of #5370

---

### Meta-Review · Area_Chair_655o · 2023-12-15

**Metareview:**

The submission introduces a method for detecting hallucinations in language models, by generating multiple answers to the same question, and then measuring the consistency of the model's hidden states. Detecting hallucination is clearly an extremely important problem. Reviewers thought that the proposed method was innovative, interesting and effective, and the paper was clearly written. In my view, the main concerns raised (e.g. only evaluating in QA settings) have been addressed in the author response and revision. Remaining limitations (e.g. requiring access to internal states) are not too problematic. Therefore, I recommend acceptance to ICLR.

**Justification For Why Not Higher Score:**

Not sure how widely used this will be in practise

**Justification For Why Not Lower Score:**

Tackles important problem with innovative and effective method.

---

### Decision · Program_Chairs · 2024-01-16

Accept (poster)